# The Influence of Typhoon-Induced Wave on the Mesoscale Eddy

**Zeqi Zhao** [1], **Jian Shi** [1,*], **Weizeng Shao** [2,*], **Ru Yao** [2] and **Huan Li** [3]

1   College of Meteorology and Oceanography, National University of Defense Technology, Changsha 410073, China; zhaozeqi22@nudt.edu.cn
2   College of Marine Sciences, Shanghai Ocean University, Shanghai 201306, China; m210200552@st.shou.edu.cn
3   National Marine Data and Information Service, Tianjin 300171, China; usher02@126.com
*   Correspondence: shijian@nudt.edu.cn (J.S.); wzshao@shou.edu.cn (W.S.);
    Tel.: +86-731-87021201 (J.S.); +86-21-61900326 (W.S.)

**Abstract:** The strong wind-induced current and sea level have influences on the wave distribution in a tropical cyclone (TC). In particular, the wave–current interaction is significant in the period in which the TC passed the mesoscale eddy. In this study, the wave fields of Typhoon Chan-hom (2015) are hindcastly simulated using a coupled oceanic model that utilizes a nested triangle grid, i.e., the finite-volume community ocean model-simulating waves nearshore (FVCOM-SWAVE) model. The forcing wind field is composited from the European Centre for Medium-Range Weather Forecasts (ECMWF) reanalysis data and the simulation using a parametric Holland model, denoted as H-E. The open boundary fields include tide data from TPOX.5 and the hybrid coordinate ocean model (HYCOM) global datasets, including sea surface temperature (SST), sea surface salinity, sea surface current, and sea level data. The simulated oceanic parameters (e.g., the significant wave height, SWH) are validated against the measurements from the Jason-2 altimeter, yielding a root mean square error (RMSE) of 0.58 m for the SWH, a correlation (COR) coefficient of 0.94, and a scatter index (SI) of 0.23. Similarly, the simulated SSTs are compared with the remote sensing products of the remote sensing system (REMSS) and the measurements from Argos, yielding an RMSE of <0.8 °C, a COR of >0.95, and an SI of <0.04. The significant zonal asymmetry of the wave distribution along the typhoon track is observed. The Stokes drift is calculated from the FVCOM-SWAVE simulation results, and then the contribution of the Stokes transport is estimated using the Ekman–Stokes numbers. It is found that the ratio of the Stokes transport to the total net transport can reach >80% near the typhoon center, and the ratio is reduced to approximately <20% away from the typhoon center, indicating that Stokes transport is an essential aspect in the water mixing during a TC. The mesoscale eddies are detected by the sea level anomalies (SLA) fusion data from AVISO. It is found that the significant wave heights, Stokes drift, and Stokes transport inside the eddy area were higher than those outside the eddy area. These parameters inside the cold mesoscale eddies were higher than t inside the warm mesoscale eddies. Otherwise, the SST mainly increased within the cold mesoscale eddies area, while decreased within the warm mesoscale eddies area. The influence of mesoscale eddies on the SST was in proportion to the eddy radius and eddy EKE.

**Keywords:** stokes transport; sea surface temperature; FVCOM-SWAVE; eddy; Typhoon Chan-hom (2015)

## 1. Introduction

Sea surface waves with a huge amount of kinetic energy are a common phenomenon in the ocean and play an important role in the heat energy exchange at the air-sea interface. Moreover, waves influence the water mass transport and water temperature in the marine mixed layer through wave–current interactions, especially on the continental shelf [1]. Consequently, waves are a critical determinant of oceanic dynamics in the field of ocean engineering. Currently, accurate waves in coastal waters have been measured using moored buoys and remote sensing satellites in nearly real-time, i.e., the National Data Buoy

Center (NDBC) of the National Oceanic and Atmospheric Administration (NOAA) [2], altimeters [3], the Surface Wave Investigation and Monitoring (SWIM) instrument onboard the Chinese-French Oceanography Satellite (CFOSAT) [4], and synthetic aperture radar (SAR) [5]. The significant wave height measurements observed by the Jason-2 altimeter have been quality controlled [6]. Moreover, the altimeter wave data are also applicable for regional research at the basin scale [7]. Reliable sea surface temperature data can also be observed using buoys and remote sensing techniques, i.e., Argos and remote-sensed data from the remote sensing system (REMSS). The primary goal of Argo is to create a systematic global network of profiling floats that provides freely available temperature and salinity data from the upper 2000 m of the ocean with global coverage. The data are available within 24 h of collection for use in a broad range of applications [8]. The REMSS dataset merges both microwave and infrared data using optimal interpolation. In addition, the product applies a diurnal model to account for day–night differences [9]. However, extreme sea states, i.e., those during a tropical cyclone (TC), are rarely measured by moored buoys due to the strong wind-induced risk. With the improvement in computational efficiency, numerical modeling is an advanced oceanography research technique. At present, several numerical models have been developed, i.e., the wave mode (WAM) [10] for sea surface waves and the Princeton ocean model (POM) [11] for ocean circulation.

At present, the third-generation wave models inheriting the principle of the WAM mainly include the WAVEWATCH-III (WW3) model developed by the National Centers for Environmental Prediction (NCEP) [12,13] and the simulating waves nearshore (SWAN) model developed by the Delft University of Technology [14,15], which are commonly used to forest and hindcast waves in regional seas [16,17] and polar regions [5,18]. In particular, it has been found that both WW3 and SWAN models have good performances in wave simulation [19,20] under TC conditions. The forcing wind field is an important source while simulating the wave field, and the extremely huge wave induced by a typhoon would be severely underestimated if only the reanalysis wind products (i.e., the Climate Forecast System Reanalysis (CFSR), ERA-Interim (ERA), and Cross-Calibrated Multi-Platform version 2 (CCMPV2)) were utilized. The hybrid wind fields combined with parametric typhoon models (i.e., the Holland model) and the reanalysis of wind products can improve the performance of wave models [21–24]. In the context of the Kuroshio Current and wind-induced current, the accuracy of the significant wave height of the waves simulated using WW3 is improved, including the strong current term [25]. In the literature, TC-induced waves are generated in deep water and then propagate to the continental shelf in the complex nearshore environment. During the TC landing period, changes in the terrain and bottom friction can result in a storm surge distinguished by a huge sea level rise and tide current. The interaction between the waves and storm surge should be considered in TC wave simulations [26,27]. In the application of numerical wave models to coastal waters, the unstructured grid has the capability to illustrate the shorelines, ensuring the variability of the water depth from several kilometers to hundreds of meters. Coastal ocean circulation models, i.e., the finite-volume community ocean model (FVCOM) [28] and the SWAN model, share a unique unstructured grid. Moreover, the FVCOM can be used to modulate the SWAN model, herein referred to as the FVCOM-SWAVE model, allowing examination of the interactions between the various dynamics [29,30] and waves.

The effects of ocean waves on the mixing layer mainly include wave breaking, radiation stress, non-breaking waves, and Stokes drift [31,32]. The small-scale dynamic processes, i.e., wave breaking, radiation stress, and non-breaking waves under extreme conditions, have been widely discussed [33]. It is recognized that the Stokes drift [34,35] is the consequent emergence of an additional flow in the direction of the wave motion. As a result, the shear change in the distribution of the sea surface current has significant influences on the sea temperature and salinity distributions in the upper ocean mixing layer [36,37]. Moreover, it has been recognized in recent studies [38] that Stokes drift plays an important role in global large-scale ocean circulation. This behavior is extraordinary due to the huge waves and wave-induced Stokes drift on the sea surface under extreme

meteorological conditions [39,40]. To better understand the response of the ocean to a TC, Stokes drift is constructive in grasping the upper ocean dynamic mechanism during a TC. The research on the Stokes drift induced by TC waves is still open to discussion because the influence of the depth scales and the contribution of Stokes transport to the total net transport are unclear. The mesoscale eddies, including cyclonic and anticyclonic eddies, can generate mass and heat transport in the upper layer of the ocean, changing the current direction, SST, and so on. This ocean phenomenon is very common and can develop into a typhoon. It is generated by the typhoon, and the interaction of the mesoscale eddies and the TC waves is a noteworthy issue. To further deal with the above-mentioned issues, the characteristics of the currents and waves during Typhoon Chan-hom (2015) were studied. The track of the above typhoon, which occurred in the China Sea, is illustrated in Figure 1.

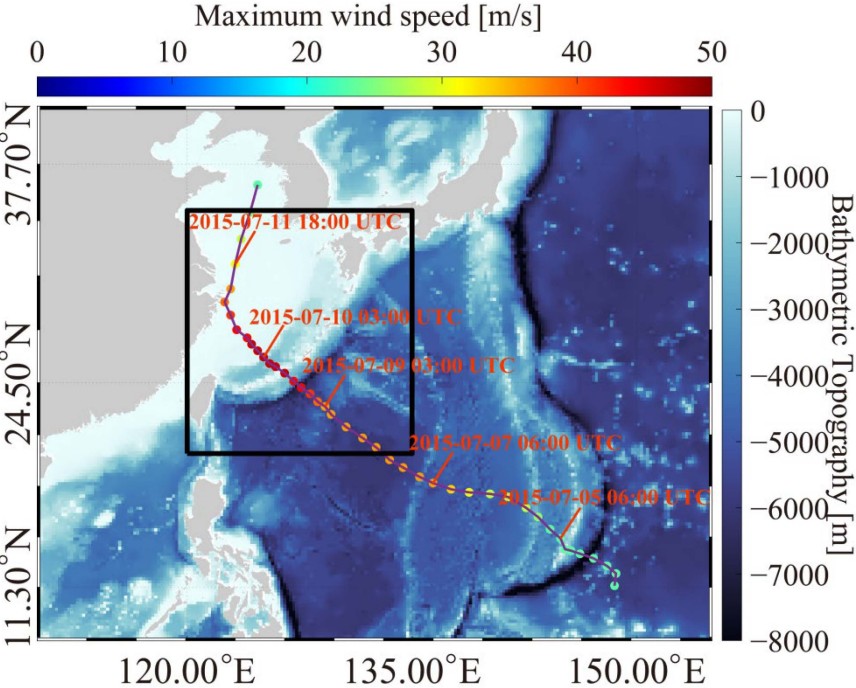

**Figure 1.** Tracks of Typhoon Chan-hom overlaid by the water depth data from the General Bathymetry Chart of the Oceans (GEBCO). The colored dots represent the maximum wind speeds during the typhoon periods. The black square represents the study area.

In previous studies, the current and sea level simulated using FVCOM were treated as open boundary conditions in wave simulations using SWAN [11]. In particular, the combination of the FVCOM and SWAN models was used for studying the effects of sea level rise on storm surges and waves within the Yangtze River Estuary [41]. In this study, the current and wave fields during Typhoon Chan-hom (2015) were simulated using the FVCOM-SWAVE model, forced by a reconstructed wind field composited from the parametric Holland model [42] and high-resolution European Center for Medium-Range Weather Forecasts (ECMWF) wind data. Chen et al. [43] evaluated the ADCIRC-SWAN, FVCOM-SWAVE, and SELFE-WWM for simulating extratropical storms and found that the FVCOM-SWAVE [44] could simulate large, significant wave heights on the coasts concerning the current-wave interaction using a second-order accurate upwind finite-volume scheme. Otherwise, the mesoscale eddies were well detected using the sea level anomalies (SLA) fusion data from AVISO [45], especially at moderate latitudes such as the China sea. The remainder of this paper is organized as follows. Section 2 describes the datasets and modeling settings of the FVCOM. In addition, the methods of calculating the Stokes drift and detecting mesoscale eddy are presented. In Section 3, the simulated waves are validated against observations from the Jason-2 altimeter, and the simulated sea surface temperature (SST) is compared with measurements from Argos and daily SST fusion data.

In addition, the mesoscale eddy, wave distribution, Stokes drift to the total current, and Stokes transport are depicted in this section. The SST cooling and the inter-relationship of the eddy, waves, and SST are discussed in detail in Section 4, and the conclusions are summarized in Section 5.

## 2. Materials and Methods

In this section, a description of the typhoon case studies is briefly presented. Then, the basic principles of the FVCOM-SWAVE and the settings of the model are described. In addition, the method of calculating the Stokes drift is presented.

### 2.1. Typhoon Cases

In this study, Typhoon Chan-hom, which passed over the Northwest Pacific from 29 June to 12 July 2015, was chosen as a case study. The location and maximum wind speed data for Typhoon Chan-hom were collected from the Tokyo-Typhoon Center of the Japan Meteorological Agency (JMA). Figure 1 shows the track of Typhoon Chan-hom overlaid with the water depth data from the General Bathymetry Chart of the Oceans (GEBCO). The colored dots represent the maximum wind speeds during the typhoon periods, and the black square represents the study area. This area covers latitudes of 20–35° N and longitudes of 120–135° E. It can be seen that Typhoon Chan-hom crossed the open ocean and nearshore area, which contained complex ocean phenomena, i.e., typhoon-induced currents and changes in the sea-water level. This typhoon behavior provides us with the opportunity to investigate the wave–current interaction effect on the typhoon wave simulation using the FVCOM-SWAVE model.

### 2.2. Settings of the FVCOM-SWAVE

FVCOM is a 3D prognostic, unstructured grid, finite volume coastal ocean model [28], which has been widely used for studies of the hydrodynamics of estuaries and coasts worldwide. The name of FVCOM has been changed to the finite volume community ocean model to extend it to more features. The model contains both Cartesian and spherical coordinates. In the horizontal direction, the discrete methods are triangle grids, and the turbulence closure is the Mellor and Yamada [46] level 2.5 (MY-2.5) scheme. In the vertical direction, the discrete method is the generalized terrain-following coordinate system, including the commonly used sigma coordinates, and the turbulence closure is the Smagorinsky scheme. FVCOM includes a wet-or-dry treatment, making it suitable for large tidal flat areas.

Qi et al. [47] modified the popular wave model SWAN code and developed the FVCOM-SWAVE model, which is a very practical and significant improvement for the application of FVCOM. The primitive momentum equations, T/S equations, and the continuity equation of the FVCOM-SWAVE model from Qi et al. [46] and Wu et al. [48] are adopted in this paper.

$$\frac{\partial uD}{\partial t} + \frac{\partial u^2 D}{\partial x} + \frac{\partial uvD}{\partial y} + \frac{\partial u\omega}{\partial \sigma} - fvD = -D\frac{\partial(g\eta + p_{atm})}{\partial x} - D\int_{\sigma}^{0}\left(D\frac{\partial b}{\partial x} - \sigma\frac{\partial D}{\partial x}\frac{\partial b}{\partial \sigma}\right)d\sigma$$
$$-\left(\frac{\partial DS_{xx}}{\partial x} + \frac{\partial DS_{xy}}{\partial y}\right) + \sigma\left(\frac{\partial D}{\partial x}\frac{\partial S_{xx}}{\partial \sigma} + \frac{\partial D}{\partial y}\frac{\partial S_{xy}}{\partial \sigma}\right) + \frac{\partial \tau_x}{\partial \sigma}, \tag{1}$$

$$\frac{\partial vD}{\partial t} + \frac{\partial uvD}{\partial x} + \frac{\partial v^2 D}{\partial y} + \frac{\partial v\omega}{\partial \sigma} + fuD = -D\frac{\partial(g\eta + p_{atm})}{\partial y} - D\int_{\sigma}^{0}\left(D\frac{\partial b}{\partial y} - \sigma\frac{\partial D}{\partial y}\frac{\partial b}{\partial \sigma}\right)d\sigma$$
$$-\left(\frac{\partial DS_{xy}}{\partial x} + \frac{\partial DS_{yy}}{\partial y}\right) + \sigma\left(\frac{\partial D}{\partial x}\frac{\partial S_{xy}}{\partial \sigma} + \frac{\partial D}{\partial y}\frac{\partial S_{yy}}{\partial \sigma}\right) + \frac{\partial \tau_y}{\partial \sigma}, \tag{2}$$

$$\frac{\partial Du}{\partial x} + \frac{\partial Dv}{\partial y} + \frac{\partial \omega}{\partial \sigma} + \frac{\partial \eta}{\partial t} = 0, \tag{3}$$

$$\frac{\partial \theta D}{\partial t} + \frac{\partial \theta uD}{\partial x} + \frac{\partial \theta vD}{\partial y} + \frac{\partial \theta \omega}{\partial \sigma} = \frac{1}{D}\frac{\partial}{\partial \sigma}\left(K_h\frac{\partial \theta}{\partial \sigma}\right) + D\hat{H} + DF_\theta, \tag{4}$$

$$\frac{\partial sD}{\partial t} + \frac{\partial suD}{\partial x} + \frac{\partial svD}{\partial y} + \frac{\partial s\omega}{\partial \sigma} = \frac{1}{D}\frac{\partial}{\partial \sigma}\left(K_h\frac{\partial s}{\partial \sigma}\right) + DF_s, \tag{5}$$

$$\rho = \rho(\theta, s), \tag{6}$$

where $x$, $y$, and $\sigma$ are the east, north, and vertical sigma axes; $u$, $v$, and $\omega$ are the velocity components in the $x$, $y$, and $\sigma$ directions; $h$ is the mean water depth; $D = h + \eta$ is the total water depth; $\eta$ is the free surface elevation; $\rho$ is the density; $b$ is buoyancy; $f$ is the Coriolis coefficient; $p_{atm}$ is the atmospheric pressure; $\tau_x$ and $\tau_y$ denote the wind stress in the $x$ and $y$ directions; $t$ and $s$ are the temperature and salinity, respectively; $\hat{H}$ is the source and sink of the heat flux; $K_h$ is the vertical eddy diffusivity; $F_\theta$ and $F_s$ are the horizontal diffusivity; $S_{xx}$, $S_{xy}$, and $S_{yy}$ are the radiation stress, and more details can be found in Qi et al. [47].

The wave action balance equation in FVCOM-SWAVE is as follows:

$$\frac{\partial N}{\partial t} + \nabla \cdot \left[\left(\vec{C_g} + \vec{V}\right)N\right] + \frac{\partial C_\sigma N}{\partial \sigma} + \frac{\partial C_\theta N}{\partial \theta} = \frac{S_{tot}}{\sigma}, \tag{7}$$

where $N$ is the wave action density spectrum; $t$ is time; $C_g$ is the wave group velocity; $V$ is the ambient water velocity; $\sigma$ and $\theta$ are the relative frequency and wave direction; $C_\sigma$ and $C_\theta$ are the wave velocity in the relative frequency and wave direction, respectively; and $S_{tot}$ is the source and sink term, including the effects of wind-induced wave growth, three-wave interactions, four-wave interactions, white capping, bottom friction, and wave breaking. Following the discrete approach used in FVCOM-SWAVE, Equation (7) is divided into four equations as follows:

$$\frac{N^{n+\frac{1}{4}} - N^n}{\Delta t} + \frac{\partial(C_\sigma N)}{\partial \sigma} = 0, \tag{8}$$

$$\frac{N^{n+\frac{2}{4}} - N^{n+\frac{1}{4}}}{\Delta t} + \frac{\partial(C_\theta N)}{\partial \theta} = 0, \tag{9}$$

$$\frac{N^{n+\frac{3}{4}} - N^{n+\frac{2}{4}}}{\Delta t} + \nabla \cdot \left[\left(\vec{C_g} + \vec{V}\right)N\right] = 0, \tag{10}$$

$$\frac{N^{n+1} - N^{n+\frac{3}{4}}}{\Delta t} = \frac{S_{tot}}{\sigma}, \tag{11}$$

where $n$ denotes the nth time step, and $\Delta t$ is the time interval for the numerical integration. Equations (8) and (9) describe the change in action density spectrum in spectral space, which are solved by the flux-corrected transport method (FCT) and the Crank–Nicolson method, respectively. Equation (5) describes the propagation of the waves in geographic space, which is solved by either an explicit finite-volume upwind advection scheme (directly adopted from FVCOM) or a semi-implicit finite-volume upwind advection scheme. Equation (6) is the growth, transfer, and decay of the waves driven by the source terms. It is solved by a semi-implicit integration scheme as used in the WAM model and the WAVEWATCH III model [47].

In this paper, an unstructured triangular horizontal grid is used, which consists of 376,298 compute nodes and 732,546 triangular grids. The land boundary, island boundary, and ocean open boundary are all taken into account. A sponge layer is specified around this area with a damping zone weighted from the open boundary into the interior with a specified influence radius. A total of 45 sigma layers are used in the vertical. The internal time step used in this study is 5 s, with an internal-external mode split ratio of 10, and the output time interval is 1 h. The maximum wind speeds from ECMWF are less than those from the JMA best track data. Thus, the H-E winds (the forcing wind field composited from the European Centre for Medium-Range Weather Forecasts (ECMWF) reanalysis data and simulated winds using a parametric Holland model), denoted as H-E, were used as

the forcing wind field [42,44]. While building the H-E winds, the large values between the ECMWF wind data and simulated Holland winds were chosen to obtain H-E wind fields. The tide data from TPXO.5 and the SST, salinity, and current from the hybrid coordinate ocean model (HYCOM) were treated as the open boundary conditions that allow the momentum or mass to be radiated out of or flow into the computational domain. It should be noted that the elapsed time of the model was from 15 June to 20 July 2015 for Typhoon Chan-hom.

As an example, the unstructured grid of the study area is shown in Figure 2. The sea surface salinity and temperature data from HYCOM during Typhoon Chan-hom at 00:00 UTC on 9 July 2015 are shown in Figure 3. Figure 4 shows the sea surface current speed from HYCOM during Typhoon Chan-hom at 00:00 UTC on 9 July 2015, with the position of Argo. At present, the soil moisture active passive (SMAP) instrument has the ability to measure 0.25° gridded sea surface wind vectors with a swath coverage of 350 km and a maximum wind speed of up to 40 m/s during typhoon conditions [49]. The SMAP radiometer winds are compared with the winds from other satellites and numerical weather models for validation. The root mean square difference (RMSD) with WindSAT is 1.7 m/s below 20 m/s wind speeds. There is also a good agreement with the airborne stepped-frequency radiometer wind speeds with an RMSD of 4.6 m/s for wind speeds in the range of 20–40 m/s [50]. Figure 5a shows the wind speed from the H-E winds during Typhoon Chan-hom at 00:00 UTC on 6 July 2015. There are more than 100,000 matchups from SMAP crossing the simulation area during Typhoon Chan-hom from 29 June 2015 to 8 July 2015. A comparison of the measurements from SMAP and the wind speed from the H-E wind is shown in Figure 5b, yielding a root mean square error (RMSE) of 1.67 m/s, a correlation coefficient (COR) of 0.91, a regression coefficient (COEF) of 0.95, and a scatter index (SI) of 0.24. Thus, it is concluded that the H-E wind field is reliable for FVCOM-SWAVE simulations.

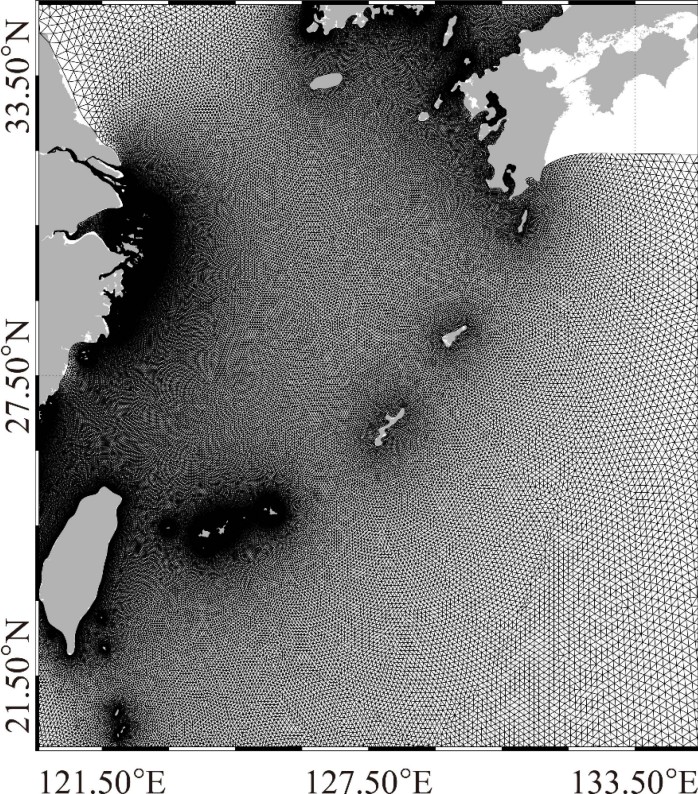

**Figure 2.** Map of the unstructured grid for the study area, which corresponds to the black square shown in Figure 1.

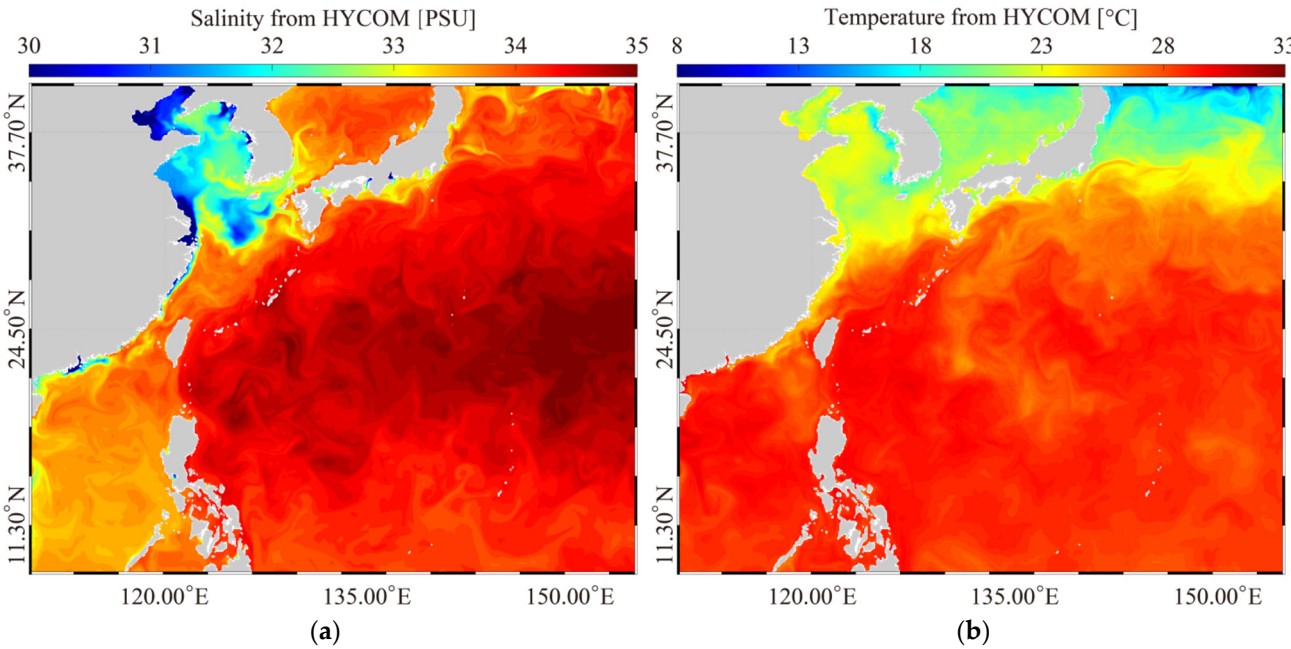

**Figure 3.** (**a**) Sea surface salinity map and (**b**) sea surface temperature (SST) map from the hybrid coordinate ocean model (HYCOM) at 00:00 UTC on 9 July 2015.

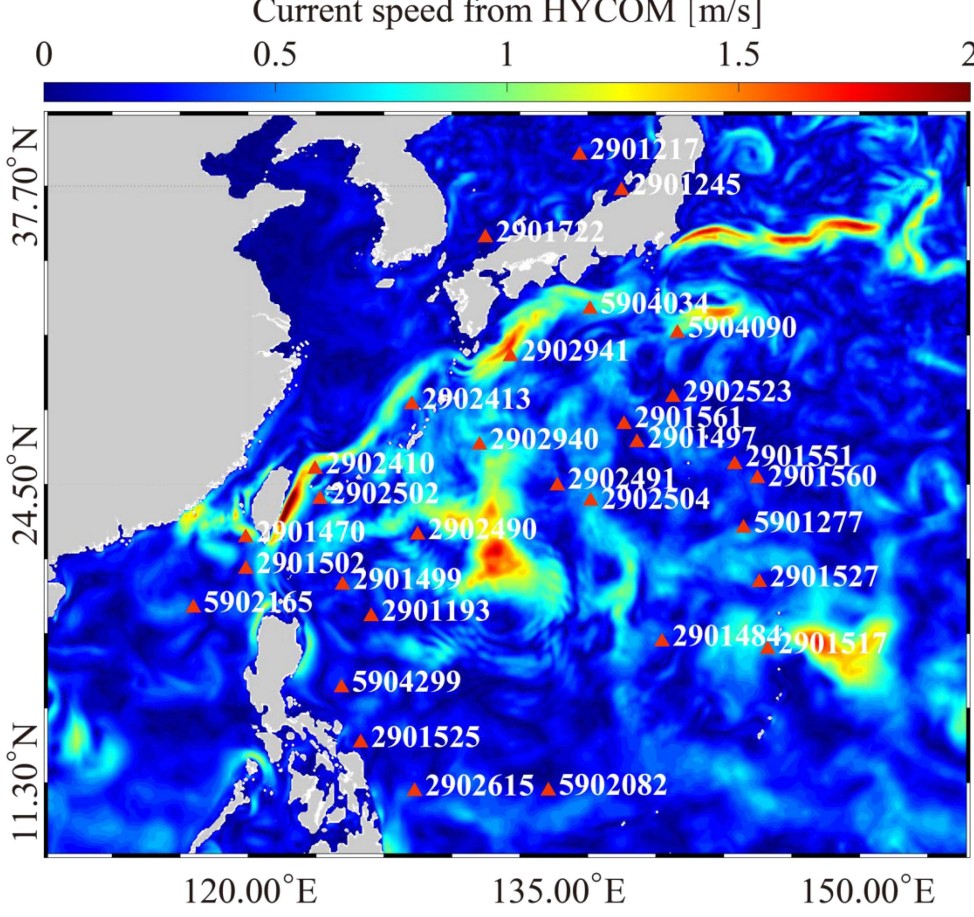

**Figure 4.** Sea surface current map from the hybrid coordinate ocean model (HYCOM) at 00:00 UTC on 9 July 2015, in which the black triangle represents the geographic location of Argos.

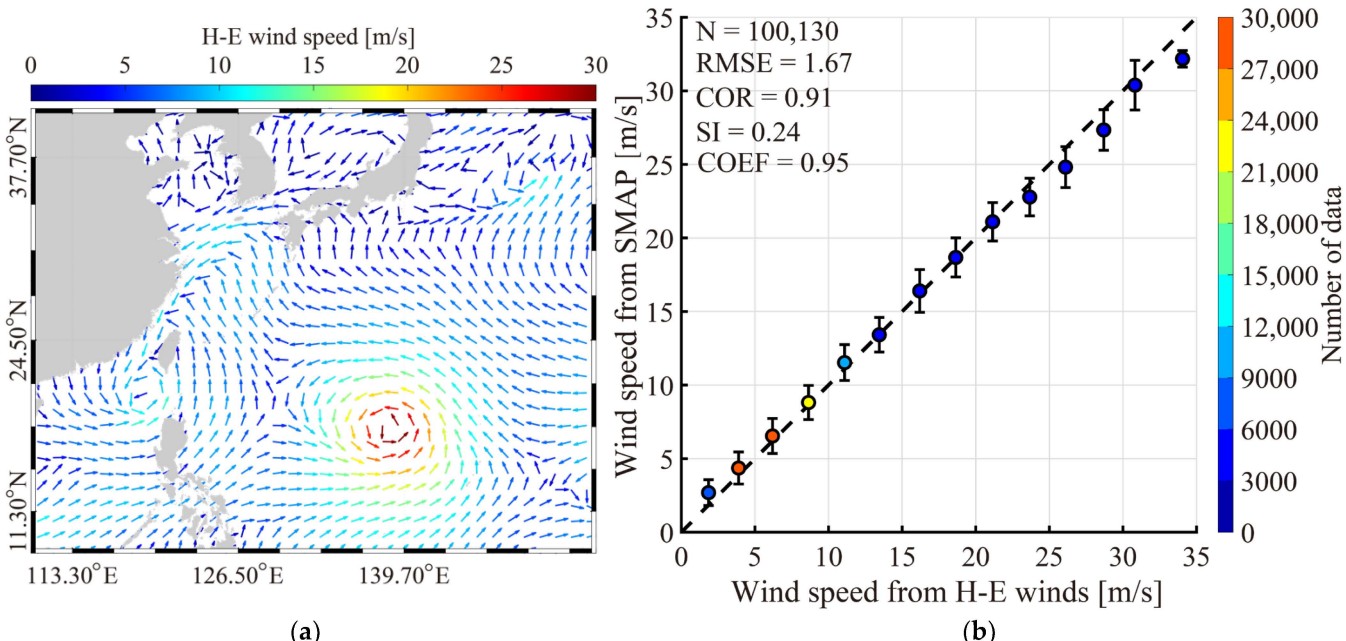

**Figure 5.** (**a**) H-E wind map during Typhoon Chan-hom at 00:00 UTC on 6 July 2015 and (**b**) a comparison of the measurements from SMAP and the wind speeds from the H-E wind field during Typhoon Chan-hom.

*2.3. Stokes Drift*

The Stokes drift caused by the single-frequency deep-water gravity waves when the fluid is non-rotational and non-viscos can be described as follows [51]:

$$\vec{u}_{st} = \vec{U}_s \exp\left(\frac{z}{d_s}\right), \tag{12}$$

where $u_{st}$ is the Stokes drift, $U_s$ is the sea surface Stokes drift, $z$ is the depth, and $d_s$ is the Stokes depth scale. The Stokes depth scale [$d_s = 1/(2k)$] is a variable influenced by the wave number $k$. When the water depth exceeds the Stokes depth, the effect of the Stokes drift becomes slight and can be ignored.

In this paper, we utilize the wave parameters simulated using the FVCOM-SWAVE model to calculate the Stokes drift on the sea surface [52].

$$\vec{u}_{st} = \vec{U}_s \exp\left(\frac{8\pi^2 z}{gT^2}\right), \tag{13}$$

$$\vec{U}_s = \frac{2\pi^3 H_s^2}{gT^3} \vec{D}, \tag{14}$$

where $H_s$ is the significant wave height, $T$ is the average period, $D$ is the wave propagation direction, and g is the gravitational acceleration.

As depicted in Equation (1), the Stokes drift moves fastest at the sea surface, and its velocity decreases exponentially with the increasing water depth. Within the Stokes depth scale, the total water particle transport induced by the Stokes drift is defined as the Stokes transport, which increases the mass and energy exchange in the upper ocean layer. The Stokes transport can be calculated by integrating the Stokes drift with the water depth as follows:

$$T_S = \int_{-d_s}^{0} \vec{u}_{st}(z) \mathrm{d}z = \pi \frac{a^2}{T} \vec{D}, \tag{15}$$

where $T_S$ is the Stokes transport, and $a$ is the wave amplitude.

The contribution of Stokes transport in the upper ocean layer can be assessed using the Ekman–Stokes number (ES).

$$ES = \frac{T_S}{T_E + T_S},\tag{16}$$

$$T_E = -\frac{z \times \vec{\tau}_0}{f\rho},\tag{17}$$

$$\vec{\tau}_0 = \rho_a C_d \left| \vec{U}_{10} \right| \vec{U}_{10},\tag{18}$$

where $T_E$ is the Ekman transport induced by the wind [53], $\tau_0$ is the sea surface wind stress, $f$ (=$2\omega\sin\varphi$, $\omega$ is the angular velocity of the Earth's rotation, $\varphi$ is the latitude) is the Coriolis parameter, $\rho$ is the density of seawater, $\rho_a$ (1.225 kg/m$^3$) is the density of air, $C_d$ is the drag coefficient [$C_d = 0.8 + 0.065 \,|\, U_{10}\,|) \times 10^{-3}$] [54], and $U_{10}$ is the wind speed 10 m above the sea surface. In this study, these wave parameters, including wave number, significant wave height, average period, and wave propagation direction, were simulated by the FVCOM-SWAVE, and the wind speed 10 m above the sea surface was the H-E wind. Furthermore, the large value of the Ekman–Stokes number indicated that the contribution of vertical eddy motion in the Ekman layer to mixing increases, and it might break the ocean-layered structure and influence the tracks and intensities of tropical cyclones.

### 2.4. Mesoscale Eddy Identification

The sea surface current field can be identified by the two-dimensional SLA data using the following equations [55]:

$$u = -\frac{g}{f}\frac{\partial Ha}{\partial y},\tag{19}$$

$$v = \frac{g}{f}\frac{\partial Ha}{\partial x},\tag{20}$$

where $Ha$ represents the SLA fields; $u$ and $v$ represent the current components at latitude $x$ and longitude $y$, respectively; the gravity acceleration constant is denoted as $g$ (=9.8 m$^2$/s); the Coriolis force constant is denoted as $f$, which is inversely proportional to latitude. Using the vector geometry algorithm, the activity intensity (*EKE*) of the mesoscale eddy can be detected by the current components as follows:

$$EKE = \frac{1}{2}(\bar{u}^2 + \bar{v}^2),\tag{21}$$

Specifically, the eddy center is defined as follows.

1.  The components of the current speed u and v on both sides of the eddy center have opposite signs and their absolute value increases with movement away from the center;
2.  The minimum velocity point within the selected range is defined as the eddy center; and
3.  The direction of the two adjacent velocity vectors around the eddy center has to be close to each other, which is located in the same or adjacent quadrants, to ensure the same direction of rotation.

After the eddy center was defined, the eddy edge was determined as the contour of the outermost closed stream function around the eddy center. In addition, as the direction of the rotation was confirmed, the warm and cold mesoscale eddies could be distinguished (i.e., warm eddies rotate clockwise and cold eddies rotate counterclockwise in the Northern Hemisphere; the rotation direction was opposite in the Southern Hemisphere). The radius of the eddy was defined as the average distance from the center to each point on the edge of the eddy.

In this study, the daily-averaged SLA fusion data collected from AVISO CNES was used to detect the mesoscale eddy. An example of the SLA data is shown in Figure 6. It

should be noted that the spatial resolution (0.25° grid) of AVISO SLA data is relatively coarse, and only the eddy with a radius > 40 km can be detected.

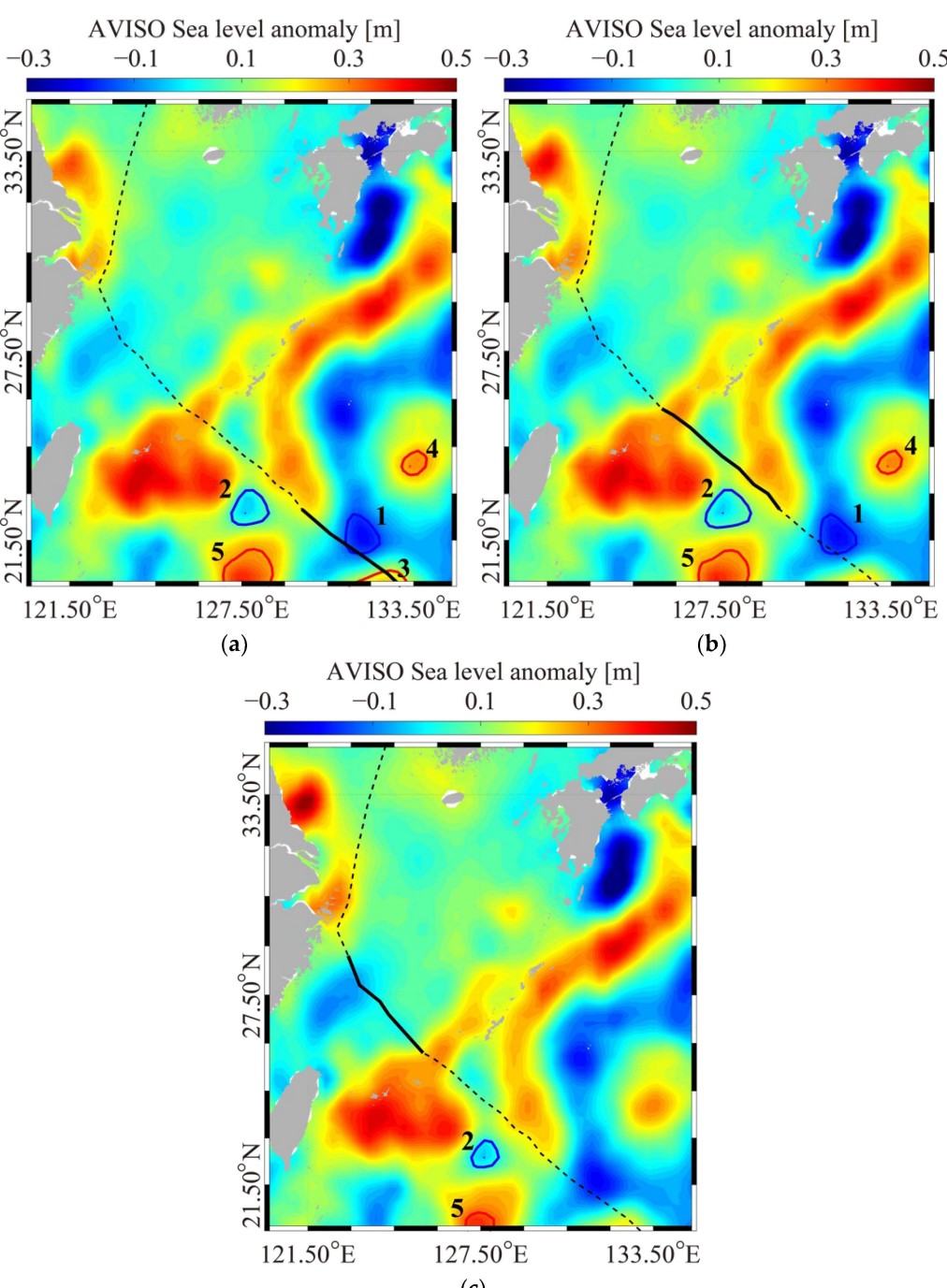

**Figure 6.** The AVISO sea level anomalies (SLA) map on (**a**) 8 July 2015; (**b**) 9 July 2015; and (**c**) 10 July 2015 with the mesoscale eddies, in which the blue and red circles represent the cyclonic and anti-cyclonic eddies, respectively. The numbers 1–5 of eddies correspond to the legend number will be mentioned in later section.

## 3. Results

In this section, the applicability of the hindcast results obtained using the FVCOM-SWAVE model is confirmed, i.e., the significant wave heights are validated against the collocated measurements from the Jason-2 altimeter, and the SST results are compared with

records from Argos and daily SST fusion data. Then, the effect of the Stokes drift produced by the waves is studied.

### 3.1. Validation of FVCOM-SWAVE Hindcast Results

To validate the significant wave heights simulated using the FVCOM-SWAVE model, significant wave height data measured from the Jason-2 altimeter were collected. There are more than 10,000 matching points across the simulation area. It should be noted that the time difference between the simulation results and the measurements from the Jason-2 altimeter is less than 15 min. RMSE, COR, and SI were used to analyze the reliability of the simulations. RMSE is used to measure the deviation between simulated values and observed values, and it is sensitive to outliers in the data. COR refers to the degree of correlation between the simulated values and the observed values. When the COR is closer to 1, it indicates a strong positive correlation between the simulated values and the observed values. SI is a measure of dispersion that describes the consistency of the simulated values. A small dispersion coefficient indicates a small degree of data dispersion. The statistical analysis yielded an RMSE of 0.58 m, a COR of 0.94, and an SI of 0.23 (Figure 7), which indicated that the FVCOM-SWAVE simulation results are consistent with the measurements from the Jason-2 altimeter (Figure 7). Comparisons of the SST simulations from the FVCOM-SWAVE model, the measurements from the Argos, and the daily fusion SST data from the remote sensing system (REMSS) are shown in Figure 8. The statistical analysis yielded an RMSE of 0.73 °C, a COR of 0.99, and an SI of 0.04 for Argos and an RMSE of 0.78 °C, a COR of 0.95, and an SI of 0.03 for the REMSS. It is concluded that the simulation results obtained using the FVCOM-SWAVE model are reliable and can be used in the subsequent study.

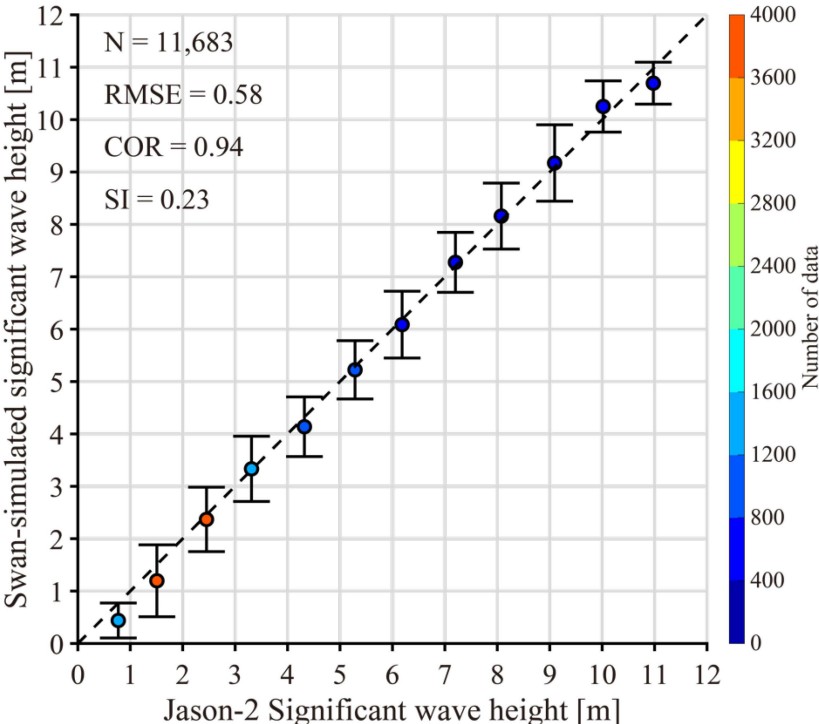

**Figure 7.** The comparison of the simulation results and measurements from the Jason-2 altimeter during Typhoon Chan-hom.

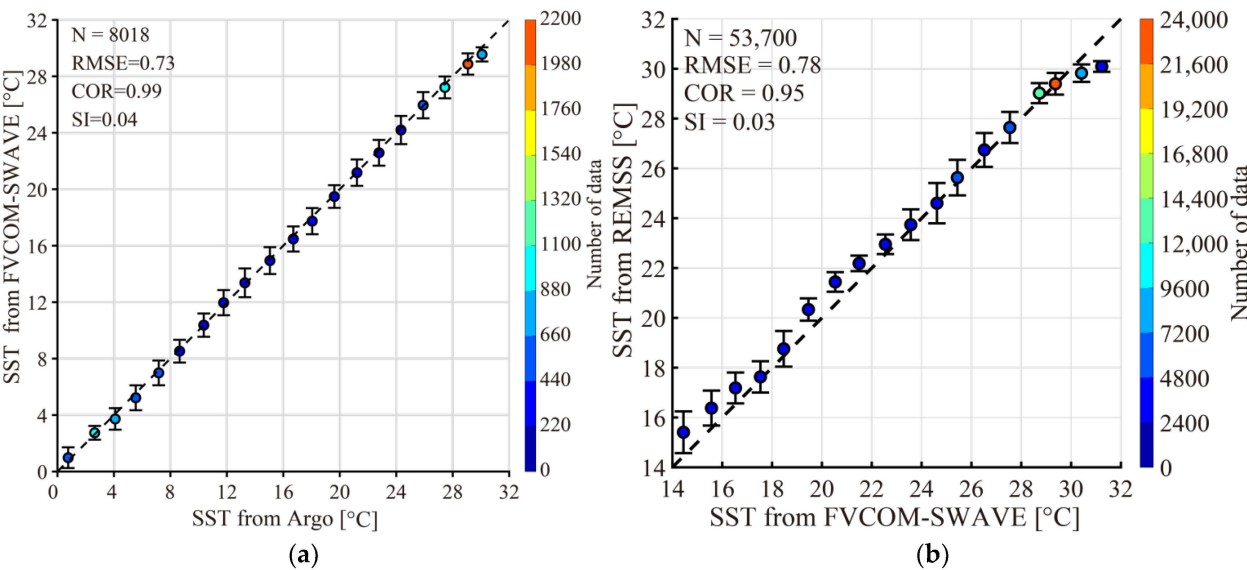

**Figure 8.** Simulated SSTs from the FVCOM-SWAVE model versus (**a**) Argos measurements and (**b**) remote sensing system (REMSS) records.

### 3.2. Spatiotemporal Distribution of Wave Simulations

Figure 9 shows the significant wave heights simulated using the FVCOM-SWAVE model during Typhoon Chan-hom. It was found that the significant wave heights were much higher in the typhoon region than in the non-typhoon region. Overall, the significant wave heights in the non-typhoon region were roughly less than 1 m, while the significant wave heights in the typhoon region reached 20 m during Typhoon Chan-hom (Figure 9a). The spatial distribution of the typhoon waves exhibits obvious asymmetry, and the high wave height area moved with the position of the typhoon. In general, significant zonal asymmetry of the wave distribution along the typhoon track was observed.

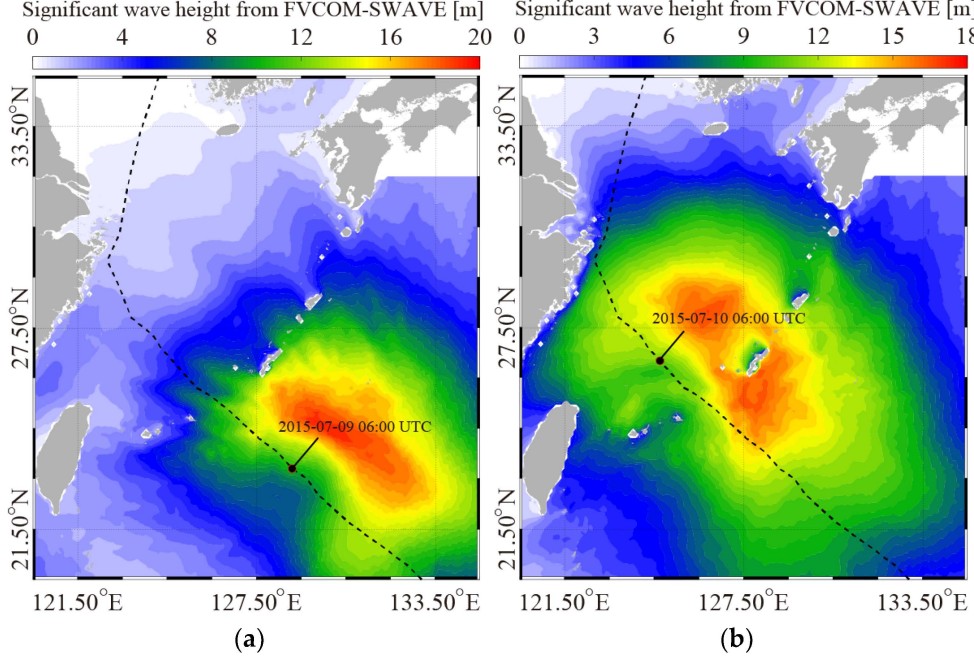

**Figure 9.** Spatiotemporal distribution of the significant wave heights simulated using the FVCOM-SWAVE model at (**a**) 06:00 UTC on 9 July 2015 during Typhoon Chan-hom; (**b**) 06:00 UTC on 10 July 2015 during Typhoon Chan-hom. The dashed line represented the typhoon track of Chan-hom, and the black point represented the corresponding typhoon position.

### 3.3. Spatiotemporal Distributions of Stokes Drift and Stokes Transport

The Stokes drift caused by the waves is generated during the typhoon period. In this study, Equation (9) was used to calculate the velocity of the Stokes drift using the FVCOM-SWAVE simulation results. The spatiotemporal distribution of the Stokes drift during Typhoon Chan-hom is presented in Figure 10. Overall, the Stokes drift in the non-typhoon region was roughly less than 0.1 m/s, while the Stokes drift in the typhoon region reached 2 m/s during Typhoon Chan-hom (Figure 10b). It can be seen that Stokes drift was generated under the typhoon conditions. In general, the spatiotemporal distribution of the Stokes drift was positively correlated with the distribution of the significant wave heights because both the wave heights and wave-induced Stokes drift were generated by the strong typhoon winds. Furthermore, the spatial distribution of the Stokes drift also moved with the position of the typhoon, which was similar to the variation in the wave heights during the typhoon.

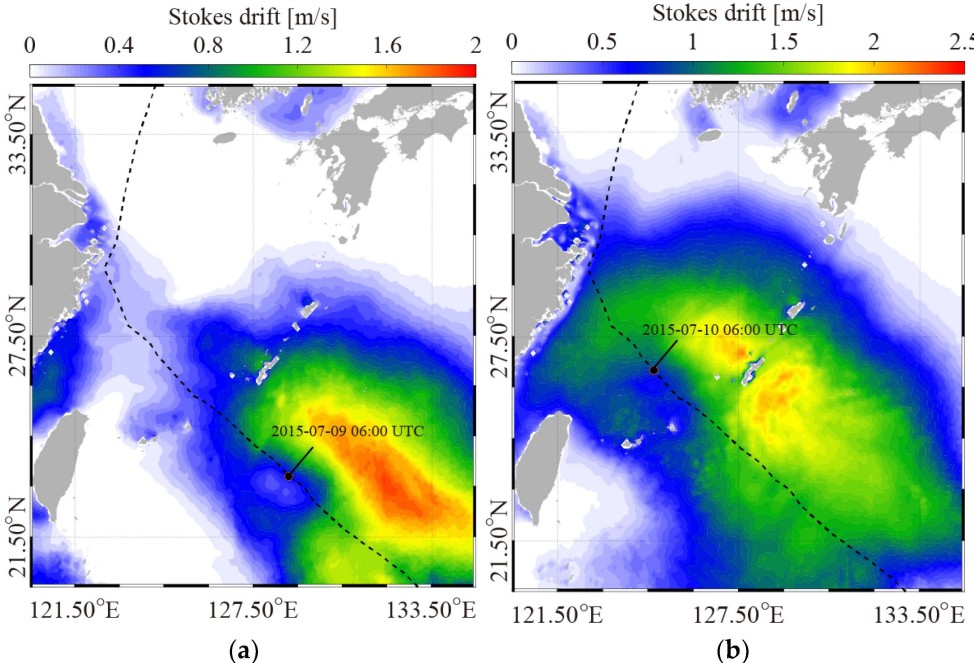

**Figure 10.** Spatiotemporal distribution of the Stokes drift at (**a**) 06:00 UTC on 9 July 2015 during Typhoon Chan-hom; (**b**) 06:00 UTC on 10 July 2015 during Typhoon Chan-hom. The dashed line represented the typhoon track of Chan-hom, and the black point represented the corresponding typhoon position.

The water transported by the Stokes drift is defined as the Stokes transport, which has a huge effect on the mass and energy exchange in the upper ocean layer. In this study, the Stokes transport induced by Typhoon Chan-hom was obtained using Equation (11). The calculated Stokes transport during the typhoon period is shown in Figure 11. It can be seen that the distributions of the Stokes transport and Stokes drift were consistent, and the strong Stokes transport region was consistent with the strong Stokes drift region. The maximum Stokes transport intensity reached 18 m$^2$/s during Typhoon Chan-hom (Figure 11a). The stronger the wind and wave field were, the greater the velocity of the Stokes drift was, and the more significant the mass and energy transport were.

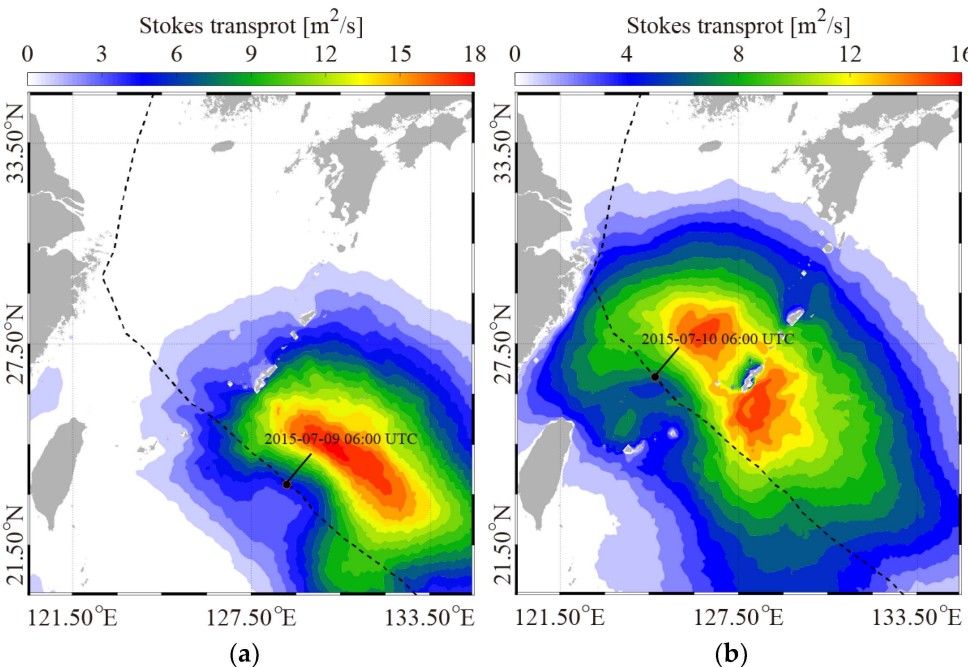

**Figure 11.** Spatiotemporal distribution of the Stokes transport at (**a**) 06:00 UTC on 9 July 2015 during Typhoon Chan-hom; (**b**) 06:00 UTC on 10 July 2015 during Typhoon Chan-hom. The dashed line represented the typhoon track of Chan-hom, and the black point represented the corresponding typhoon position.

*3.4. Spatiotemporal Distribution of Ekman–Stokes Numbers*

In order to analyze the contribution of the Stokes transport to the total transport, Equations (12)–(14) were used to calculate the Ekman–Stokes (ES) numbers. Figure 12 illustrates the spatiotemporal distribution of the ES numbers during Typhoon Chan-hom. It can be seen from Figures 10–12 that the spatial distribution of the ES is not similar to those of the Stokes drift and Stokes transport. During the influence of the typhoon, the maximum ES numbers occurred near the typhoon's eyes. Figure 12 shows that the ratio of the Stokes transport to the total net transport reached >80% near the typhoon center, but the ratio was <20% away from the typhoon center. In general, the results indicated that Stokes transport was an essential aspect of the water mixing during the TC. The Stokes drift was mainly induced by wind, and it could also be influenced by the Coriolis force and water depth. During the TC, strong wind fields were the main force of Stokes transport, which made Stokes transport dominate the water movement and mixing. The Stokes drift produces water mass movement both horizontally and vertically. The horizontal water mass movement could change the temperature, salinity, and other physical properties of the ocean surface layer. Furthermore, the increase in vertical water mixing might change the ocean layer structure and increase the depth of the mixed layer.

*3.5. Mesoscale Eddies Influence on Typhoon Waves*

Mesoscale eddies are a common marine phenomenon. Mesoscale eddies could affect the wave energy exchange and current patterns under tropical cyclone conditions by changing the vertical mixing of seawater, especially in the upper ocean layer. Otherwise, the interaction of mesoscale eddies and topography, such as submarine ridges, can also alter the path and velocity of ocean currents. These changes may affect the distribution of ocean currents throughout the entire tropical cyclone region. To study the influence of the mesoscale eddies on the typhoon waves, the mesoscale eddies detected by the SLA fusion data from 8–10 July 2015 were selected. As shown in Figure 6, two cold mesoscale eddies and three warm mesoscale eddies on 8 July 2015; two mesoscale cold eddies and two warm mesoscale eddies on 9 July 2015; and one mesoscale cold eddy and one warm mesoscale

eddy on 10 July 2105 were detected for analysis. The variation in the significant wave heights, Stokes drift, Stokes transport, and Ekman–Stokes numbers inside and outside the eddy area is illustrated in Figure 13 (black lines and green lines). As shown in Figure 13a–c, the significant wave heights, Stokes drift, and Stokes transport either increased from 8–10 July 2015 inside or outside the eddy area. Clearly, the variation in these three ocean parameters in eddy areas was more complex than in non-eddy areas, with sudden increases and decreases. The significant wave heights and Stokes transport inside the eddy area were higher than those outside the eddy area, for the mesoscale eddies enlarged the mass transport in the vertical ocean layer and changed the sea level heights. The Stokes drift inside the eddy area was smaller than that outside the eddy area before 06:00 8 July 2015 and then larger than that outside the eddy area. This was due to Typhoon Chan-hom totally moving into the study area after 06:00 8 July 2015, as illustrated in Figure 1. As for the Ekman–Stokes numbers shown in Figure 13d, with the influence of the Kuroshio and the barriers of the islands, the difference inside and outside the eddy areas was not obvious.

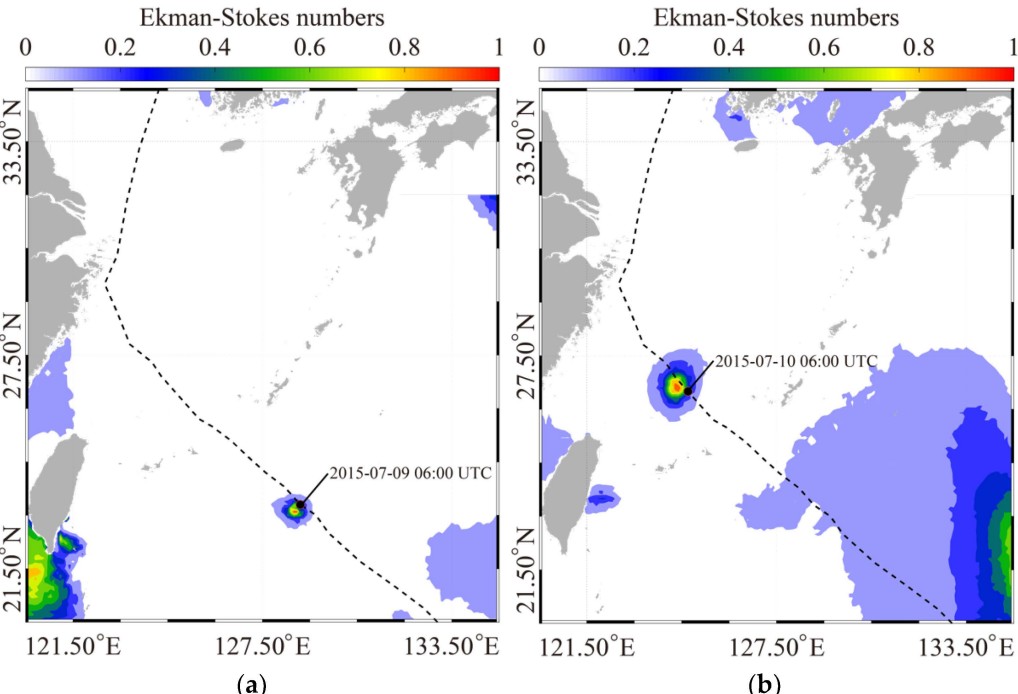

**Figure 12.** The spatiotemporal distribution of the Ekman–Stokes (ES) numbers at (**a**) 06:00 UTC on 9 July 2015 during Typhoon Chan-hom; (**b**) 06:00 UTC on 10 July 2015 during Typhoon Chan-hom. The dashed line represented the typhoon track of Chan-hom, and the black point represented the corresponding typhoon position.

To further study the different influences of the cold and warm mesoscale eddies on the typhoon waves, the variation in the significant wave heights, Stokes drift, Stokes transport, and Ekman–Stokes numbers within the cold mesoscale eddy area and the warm mesoscale eddy area is also illustrated in Figure 13 (red lines and blue lines). The cold mesoscale eddy area includes all the cold mesoscale eddy detected in the study area, which changes with the generation and dissipation of the cold mesoscale eddy, and the same for the warm mesoscale eddy area. The values of the significant wave heights, Stokes drift, and Stokes transport inside the cold mesoscale eddies areas were all greater than those inside the warm mesoscale eddies areas. As the cold eddies were cyclonic eddies, this kind of eddies pumped up the water in the lower ocean layer to the sea surface, forming the upwelling and increasing the significant wave heights. While the significant wave heights increased, the Stokes drift and Stokes transport increased. As for the warm eddies, this kind of eddie was anti-cyclonic and performed conversely compared to the cold eddies.

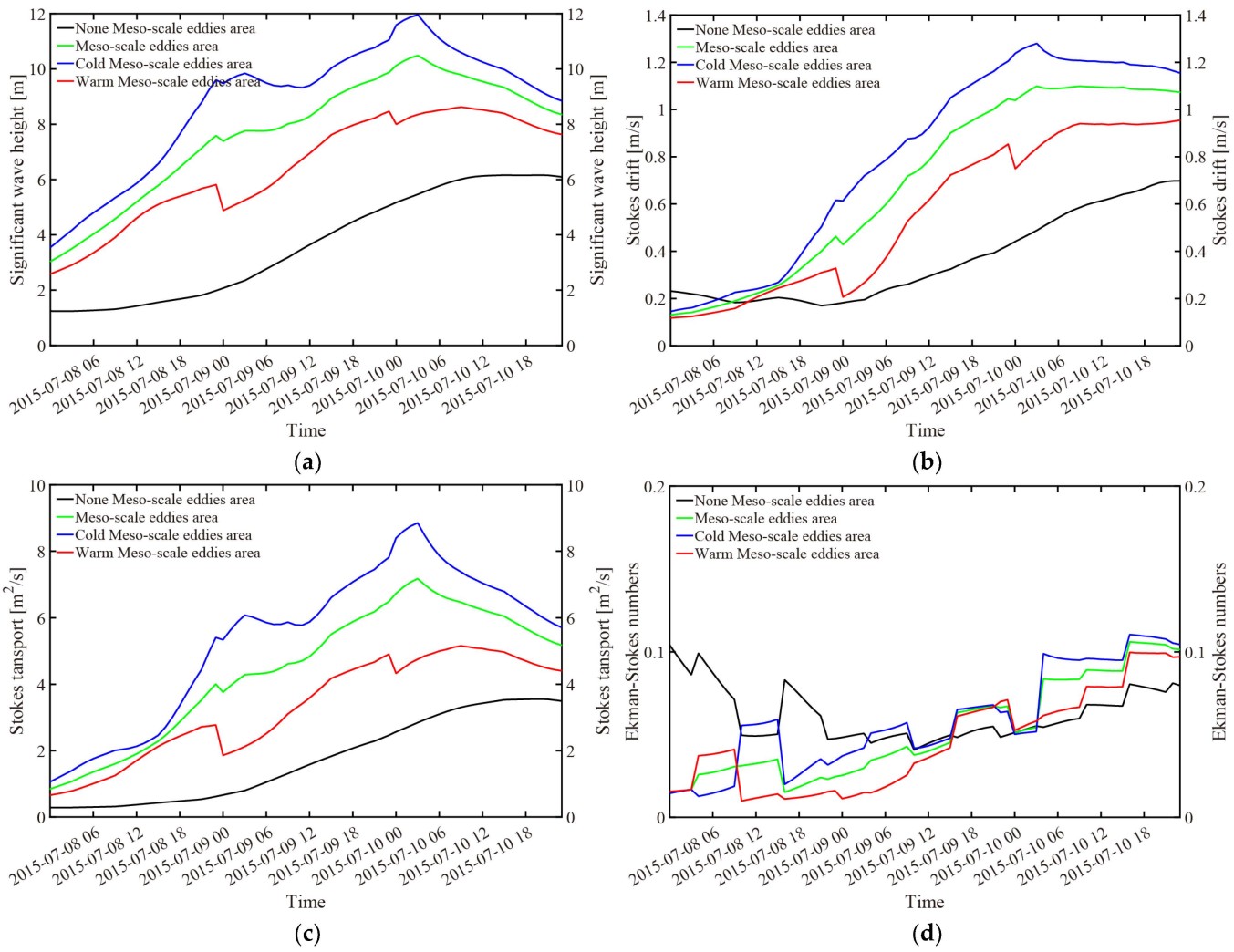

**Figure 13.** The variation in the (**a**) significant wave heights, (**b**) Stokes drift, (**c**) Stokes transport, and (**d**) Ekman–Stokes numbers during Typhoon Chan-hom from 8–10 July 2015. The black lines represent the parameters outside the eddies area, the green lines represent the parameters inside the eddies area, the blue lines represent the parameters inside the cold eddies area, and the red lines represent the parameters inside the warm eddies area.

## 4. Discussion

At present, many satellite observations [56,57] and numerical models [17] have been used to study typhoon-induced SST cooling in the open sea. In this section, the SST simulated from the FVCOM-SWAVE model was used to analyze the typhoon-induced SST cooling. The spatiotemporal distribution of the SST cooling during Typhoon Chan-hom was presented in Figure 14, in which the black lines represented the typhoon durations and the dash lines represented the typhoon tracks. It can be seen that the reduction in the SST is determined by the maximum value of −5 °C for Typhoon Chan-hom. The amplitude of the reduction varied during Typhoon Chan-hom under the effect of the Kuroshio Current. The main SST cooling occurred on the left side of Typhoon Chan-hom when the typhoon crossed the Kuroshio Current.

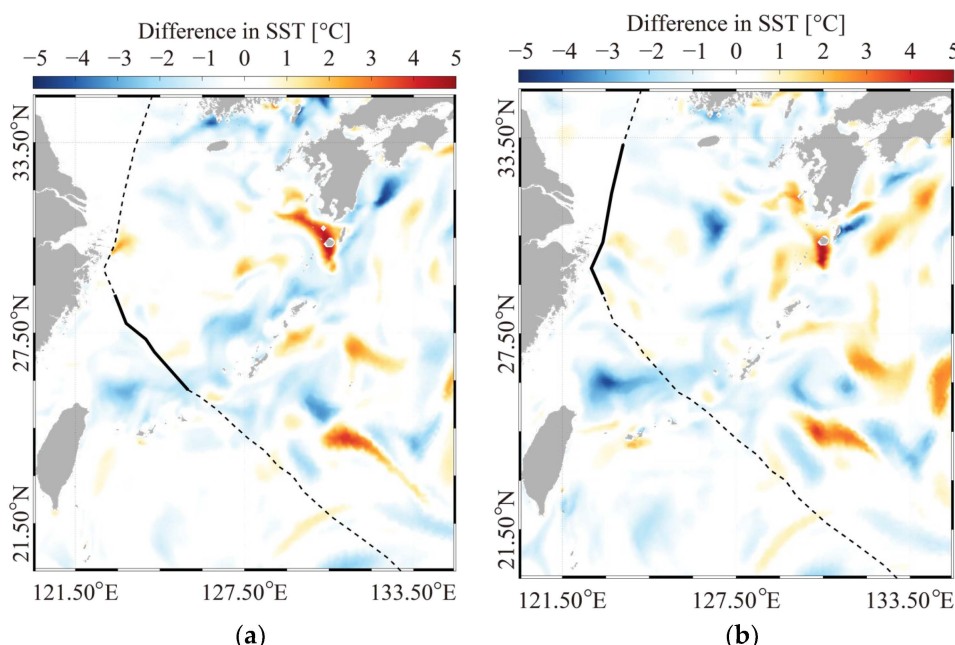

**Figure 14.** Spatiotemporal distribution of the SST cooling: (**a**) results for 10 July minus results for 8 July during Typhoon Chan-hom; (**b**) results for 11 July minus results for 9 July during Typhoon Chan-hom. The black lines represent the typhoon durations, and the dashed lines are the typhoon tracks.

In order to analyze the interaction of the mesoscale eddies and SST during Typhoon Chan-hom, the mesoscale eddies detected from 8–10 July 2015 were selected. As illustrated in Figure 15, the SST outside the mesoscale eddies (black line) gradually decreased with a value of about 0.2 °C, which was induced by the typhoon. The SST inside the mesoscale eddies (green, red, and blue lines) varied relatively complexly, with a value greater than 0.5 °C. Within the cold mesoscale eddies (blue line) area, the SST mainly increased. Conversely, within the warm mesoscale eddies (red line) area, the SST mainly decreased. This opposite performance was induced by the different physical mechanisms of the cold and warm mesoscale eddies. The cold mesoscale eddies rotated counter-clockwise, which conflicts with the typhoon and consumes energy, resulting in further weakening of the SST cooling induced by the typhoon. As for the warm mesoscale eddies, they rotated clockwise, which promoted the development of the typhoon and the SST cooling induced by the typhoon. This process was very complex and needed further study. Furthermore, as for the warm meso-scale eddy, the water in the upper ocean layer was transported to the lower ocean layer and formed the downwelling in the center of the eddy, which prevented heat transport from the lower ocean layer. Otherwise, the sudden change of the SST on 00:00 9 July 2015 and 00:00 10 July 2015 existed, as shown in Figure 15a. This was due to the fact that the number 3 warm mesoscale eddy disappeared on 9 July 2015, the number 1 cold mesoscale eddy disappeared, and the number 4 warm mesoscale eddy disappeared on 10 July 2015.

Furthermore, to study the influence of the eddy radius (r) and eddy activity intensity (EKE) on the SST, mesoscale eddies detected on 8 July 2015 in the study area were selected. As illustrated in Figure 16a, the two cold mesoscale eddies had similar eddy radii; the SST inside the eddy with an EKE of 0.0920 (blue line) increased with the maximum variation of 0.015 °C, which was greater inside the eddy with an EKE of 0.0368 (green line). In Figure 16b, the three warm mesoscale eddies had different eddy radii and eddy EKEs; the cooling effect of the warm eddy with the minimum eddy radius and eddy EKE (blue line) on the SST was unstable. The cooling effect of the warm eddy with a radius of 159.1657 km and an EKE of 0.0878 (green line) was more obvious with a maximum variation of −0.06 °C than the warm eddy with a radius of 95.4602 km and an EKE of 0.0486 (red line). It seems that the influence of mesoscale eddies on the SST was in proportion to the eddy radius

and eddy EKE. Mesoscale eddies with large radii might have stronger structures, while large eddy EKE indicates large vorticity intensity and strong wind force. Thus, mesoscale eddies with large radii and EKEs had a more significant effect on the SST cooling induced by the typhoon. This phenomenon might affect the distribution, growth, and stability of the ecological chain of phytoplankton and animals. On a global scale, the regulation of SST by mesoscale eddies may affect large-scale ocean currents and heat distribution. Overall, the couple FVCOM-SWAVE model is suitable for conducting typhoon waves and SST simulations.

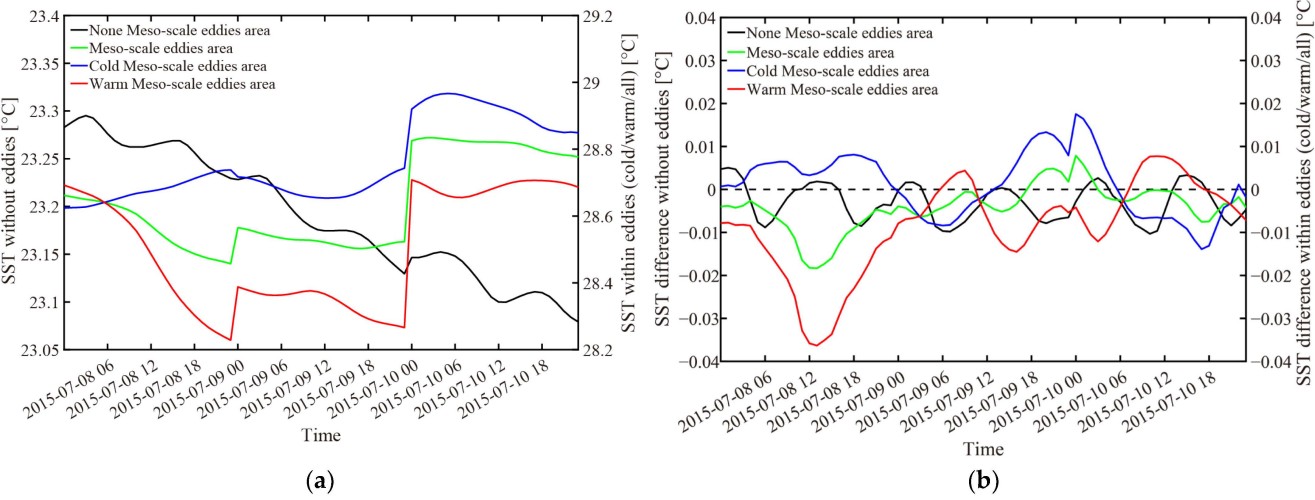

(**a**) (**b**)

**Figure 15.** The variation in the (**a**) SST and the (**b**) hourly subtractive SST difference during Typhoon Chan-hom from 8–10 July 2015. The black lines represent the parameters outside the eddies area, the green lines represent the parameters inside the eddies area, the blue lines represent the parameters inside the cold eddies area, and the red lines represent the parameters inside the warm eddies area.

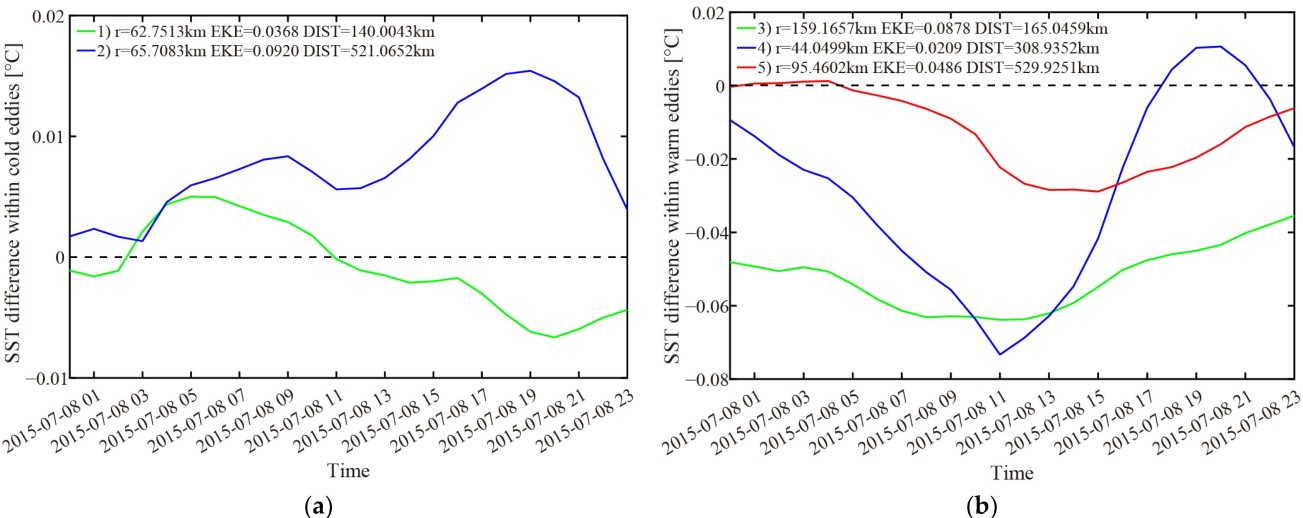

(**a**) (**b**)

**Figure 16.** The variation in the (**a**) cold mesoscale eddies and (**b**) warm mesoscale eddies during Typhoon Chan-hom on 8 July 2015. Each line presents the variation in one cold or warm eddy with different eddy radius (r), mean distance to the typhoon center (DIST), and eddy activity intensity (EKE). The number shown in legend was consistent with the number illustrated in Figure 6a.

## 5. Conclusions

Wave–current interaction is a key scientific problem in the field of oceanography. In particular, it is a challenge due to the extreme sea states during TCs. In addition, SST cooling caused by typhoon-wave pumping is a noteworthy issue [58]. Mesoscale eddies,

including cyclonic and anticyclonic eddies, can generate mass and heat transport in the upper layer of the ocean. Thus, the interaction of the SST cooling effect, typhoon-induced waves, and mesoscale eddies is worth studying. In this study, a coupled FVCOM-SWAVE model was employed to simulate the current, SST, and waves during Typhoon Chan-hom (2015). The conclusion was presented as follows:

(1) The simulated significant wave heights were validated against measurements from Jason-2, yielding an RMSE of 0.58, a COR of 0.94, and an SI of 0.23. Furthermore, the simulated SST was validated against the measurements from Argos and REMSS, yielding an RMSE of 0.73 °C/0.78 °C, a COR of 0.99/0.95, and an SI of 0.04/0.03. It is concluded that the simulation results obtained using the FVCOM-SWAVE model are reliable.

(2) The spatiotemporal distribution of the significant wave heights shows that the significant zonal asymmetry of the wave distribution occurred along the typhoon tracks. As indicated by Shi et al. [59], the stronger the typhoon, the less asymmetric the spatial distribution of SWH it causes. In addition, the asymmetry of the wind field and the topography of the shore boundary during typhoons have a significant impact on the spatial distribution of significant wave height. Thus, reliable TC winds derived from remote-sensed products are anticipated to force the numeric wave model.

(3) The simulation results obtained using the FVCOM-SWAVE model were used to calculate the Stokes drift and estimate the contribution of Stokes transport. It was found that in the open sea, the ratio of the Stokes transport to the total net transport reached 80% near the typhoon center, while it was only approximately 20% away from the typhoon center, indicating that the Stokes transport was an essential aspect of the water mixing during the TC.

(4) Using the mesoscale eddies detected by the sea level anomalies (SLA) fusion data from AVISO [60], the interaction of the SST cooling effect, typhoon-induced waves, and mesoscale eddies was analyzed. It was found that the significant wave heights, Stokes drift, and Stokes transport inside the eddy area were higher than those outside the eddy area. The values of the significant wave heights, Stokes drift, and Stokes transport inside the cold mesoscale eddies were higher than those inside the warm mesoscale eddies. Otherwise, the SST mainly increased within the cold mesoscale eddies area, while decreased within the warm mesoscale eddies area. The influence of mesoscale eddies on the SST was in proportion to the eddy radius and eddy EKE.

These understandings of Stokes transport and mesoscale eddies can be incorporated into TC simulation models to improve prediction accuracy. Furthermore, it helps improve meteorological forecasting and risk management during TCs. The wave–current–ice interaction needs to be considered by the numeric model [61] and remote sensing [62] when studying the ocean dynamics in the polar region. In the near future, the FVCOM-SWAVE model will be implemented in the Arctic Ocean, allowing for analysis of the impact of sea ice melting on ocean dynamics. In particular, the impact of waves on the SST is anticipated to be further studied.

**Author Contributions:** Conceptualization, J.S., Z.Z. and W.S.; methodology, J.S., Z.Z. and W.S.; validation, R.Y., J.S. and Z.Z.; formal analysis, J.S., W.S. and R.Y.; investigation, W.S. and H.L.; resources, H.L. and J.S.; writing—original draft preparation, W.S., H.L. and R.Y.; writing—review and editing, J.S. and Z.Z.; visualization, R.Y.; funding acquisition, W.S. All authors have read and agreed to the published version of the manuscript.

**Funding:** This research was funded by the National Key Research and Development Program of China, grant number 2023YFE0102400, the National Natural Science Foundation of China, grant numbers 41976025 and 42076238 and the Natural Science Foundation of Shanghai, grant number 23ZR1426900.

**Institutional Review Board Statement:** Not applicable.

**Informed Consent Statement:** Not applicable.

**Data Availability Statement:** Data are contained within the article.

**Acknowledgments:** We truly appreciate the provision of the Finite-Volume Community Ocean Model (FVCOM) by the Marine Ecosystem Dynamics Modeling Laboratory (MEDML). The wind field from the European Centre for Medium-Range Weather Forecasts (ECMWF) was accessed via http://www.ecmwf.int (accessed on 28 November 2023). The water depth data from the General Bathymetric Chart of the Oceans (GEBCO) were accessed via https://www.ftpstatus.com/site.php?name=edcftp.cr.usgs.gov (accessed on 28 November 2023). The TPOX.5 tide data were obtained from https://www.tpxo.net (accessed on 28 November 2023). The HYbrid Coordinate Ocean Model (HYCOM) datasets, including sea surface temperature, sea surface salinity, sea surface current, and sea-water level data, were obtained from https://www.hycom.org (accessed on 28 November 2023). The measurements from the Jason-2 altimeter and Argo data were obtained from https://data.nodc.noaa.gov (accessed on 28 November 2023).

**Conflicts of Interest:** The authors declare no conflict of interest.

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
