# Peer review of "The Influence of Typhoon-Induced Wave on the Mesoscale Eddy"

_atmosphere, doi:10.3390/atmos14121804_

Round 1

Reviewer 1 Report

Comments and Suggestions for Authors

Overall, the model design seems well-constructed. But I encourage the authors to revise the manuscript regarding the meso-scale eddies. I believe that it would be helpful for readers to understand this study if additional information was provided, such as which eddies were analyzed, the distance between the eddies and the typhoon (distance from maximum wind speed radius), and so on. After the manuscript is revised, I believe it will be a more suitable paper for this journal.

Ln 31: It can be seen that these two typhoons~. What is the two typhoons? TC Chan-hom and other?

Ln 35: Why the authors use SMAP sea surface wind vector? How about other wind products? Is there benefit that makes SMAP better than other products?

Ln 44: scatterindex > scatter index

Figure 5(b). Regression coefficient?

Section 2.4 & figure 6. Did the authors run the eddy detection to determine the eddy boundary? As the author knows, there are several criteria to define Eddie.

Figures 9-11. Please add the track of typhoon.

Section 3.4. “~ net transport reached >80% within the maximum wind radius~”. I couldn’t find maximum wind radius in the figure 12.

Section 3.5. “~mesoscale eddies~ were selected”. Which eddies were selected? Marked eddies in the figure 6?

Figure 13. Please add the distance to Rmax or typhoon center. And, what is the criteria for the non meso-scale eddy?  All eddies have the same distance relative to the typhoon center? Is it normalized?

Figure 14. I’m not sure whether this figure is suitable for showing SST cooling or not. What about SST(D+2)-SST(D-0)?

In the section 4, with the figure 15a, there is an abrupt SST increase in both warm and cold eddy at the 10 July 00:00. The authors already pointed out that there is disappearing and generation of eddies. But, I couldn’t follow it due to the limited information (such as eddy map or any?). So, it is confusing with the sentences such as; “within the cold meso~ mainly increased. Conversely, ~ decreased”. I think the variations of SST occurred on 09 July. To reduce misunderstanding, the author needs to rewrite this paragraph for clarity.

Author Response

We thank the reviewers for their constructive comments that have helped us to improve this manuscript. We provide a point-by-point reply in the document to carefully address these comments and suggestions.

Reply to Reviewer 1

General Comment: Overall, the model design seems well-constructed. But I encourage the authors to revise the manuscript regarding the meso-scale eddies. I believe that it would be helpful for readers to understand this study if additional information was provided, such as which eddies were analyzed, the distance between the eddies and the typhoon (distance from maximum wind speed radius), and so on. After the manuscript is revised, I believe it will be a more suitable paper for this journal.

Reply: Thanks for your suggestion and we added more detail information of mesoscale eddies in this study, i.e., the mesoscale eddies position illustrated in Figure 6 from July 8, 2015 to July 10, 2015 and the distance of certain mesoscale eddy to typhoon center. More detailed revisions are presented according to details.

Comment 1: Ln 31: It can be seen that these two typhoons~. What is the two typhoons? TC Chan-hom and other?

Reply: we revised the missing description in the manuscript.

Page 4, Line 1: It can be seen that Typhoon Chan-hom crossed the open ocean and nearshore area, …

Comment 2: Ln 35: Why the authors use SMAP sea surface wind vector? How about other wind products? Is there benefit that makes SMAP better than other products?

Reply: The SMAP radiometer data have a good skill to track the temporal evolution of typhoon and storm force winds. The L-band passive microwave data from the Soil Moisture Active Passive (SMAP) observatory are investigated for remote sensing of ocean surface winds during severe storms. The SMAP radiometer winds are compared with the winds from other satellites and numerical weather models for validation. The root-mean-square difference (RMSD) with WindSat or Special Sensor Microwave Imager/Sounder is 1.7 m/s below 20-m/s wind speeds. There is also a good agreement with the airborne Stepped-Frequency Radiometer wind speeds with an RMSD of 4.6 m/s for wind speeds in the range of 20-40 m/s. And the RMSD of wind direction is 18° with respect to ECMWF for wind speeds in the range of 12–30 m/s. Besides, the wind speed from ECMWF exits underestimation for high wind speed. Thus, the SMAP sea surface wind vector is a reliable data in typhoon situation.

Page 6, line 13: At present, the soil moisture active passive (SMAP) instrument has the ability to measure 0.25° gridded sea surface wind vectors with a swath coverage of 350 km and a maximum wind speed up to 40 m/s during the typhoon condition [49]. The SMAP radiometer winds are compared with the winds from other satellites and numerical weather models for validation. The root-mean-square difference (RMSD) with WindSat or Special Sensor Microwave Imager/Sounder is 1.7 m/s below 20-m/s wind speeds. There is also a good agreement with the airborne Stepped-Frequency Radiometer wind speeds with an RMSD of 4.6 m/s for wind speeds in the range of 20-40 m/s [50].

Referrence:

Meissner, T.; Wentz, F.J. The emissivity of the ocean surface between 6 and 90 GHz over a large range of wind speeds and earth incidence Angles. IEEE Trans. Geosci. Remote Sens. 2012, 50, 3004–3026.

Yueh, S.H.; Fore, A.G.; Tang, W.Q.; Hayashi, A.; Stiles, B.; Reul, N.; Weng, Y.H.; Zhang, F.Q. SMAP L-Band Passive Microwave Observations of Ocean Surface Wind During Severe Storms. IEEE Trans. Geosci. Remote Sens. 2016, 54, 7339-7350.

Comment 3: Ln 44: scatterindex > scatter index

Reply: we added the blank space in the manuscript.

Comment 4: Figure 5(b). Regression coefficient?

Reply: We calculated the regression coefficient and added it in the manuscript and figure 5(b).

Page 5, line 41-46: yielding a root mean square error (RMSE) of 1.67 m/s, a correlation coefficient (COR) of 0.91, a regression coefficient (COEF) of 0.95 and a scatter index (SI) of 0.24.

Comment 5: Section 2.4 & figure 6. Did the authors run the eddy detection to determine the eddy boundary? As the author knows, there are several criteria to define Eddie.

Reply: We added the eddy detection criteria in the manuscript.

Page 10, line 7:

Specifically, the eddy center is defined as following.

(1) The components of the current speed u and v on both sides of the eddy center have opposite signs and their absolute value increases with movement away from the center;

(2) The minimum velocity point within the selected range is defined as the indeterminacy eddy center; and

(3) The direction of the two adjacent velocity vectors around the eddy center has to be close to each other, which is located in the same or adjacent quadrants to ensure the same direction of rotation.

After the eddy center was defined, the eddy edge was determined as the contour of the outermost closed flow function around the eddy center. Besides, as the direction of the rotation was confirmed, the warm and cold mesoscale eddies could be distinguished (i.e., warm eddies rotate clockwise and cold eddies rotate counterclockwise in the Northern Hemisphere, the rotation direction was opposite in the Southern Hemisphere). The radius of the eddy was defined as the average distance from the center to each point on the edge of the eddy.

Comment 6: Figures 9-11. Please add the track of typhoon.

Reply: We added the track of Typhoon Chan-hom in the Figures 9-11.

Comment 7: Section 3.4. “~ net transport reached >80% within the maximum wind radius~”. I couldn’t find maximum wind radius in the figure 12.

Reply: We revised the description of this phenomenon in the manuscript.

Page 15, line 3: Figures 12 showed that the ratio of the Stokes transport to the total net transport reached >80% near the typhoon center, but the ratio was <20% away the typhoon center.

Comment 8: Section 3.5. “~mesoscale eddies~ were selected”. Which eddies were selected? Marked eddies in the figure 6?

Reply: We replotted figure 6, and mesoscale eddies existed on 8 July 2015; 9 July 2015 and 10 July 2015 were shown in figure 6(a)-(c) respectively. Two cold mesoscale eddies, three warm mesoscale eddies on 8 July 2015; two mesoscale cold eddies, two warm mesoscale eddies on 9 July 2015 and one mesoscale cold eddy, one warm mesoscale eddy on 10 July 2105 were selected for analysis. 

Page 15, line 24: As to study the influence of the mesoscale eddies on the typhoon waves, the mesoscale eddies detected by the SLA fusion data from July 8, 2015 to July 10, 2015 were selected. As shown in Figure 6, two cold mesoscale eddies, three warm mesoscale eddies on 8 July 2015; two mesoscale cold eddies, two warm mesoscale eddies on 9 July 2015 and one mesoscale cold eddy, one warm mesoscale eddy on 10 July 2105 were detected for analysis.

Comment 9: Figure 13. Please add the distance to Rmax or typhoon center. And, what is the criteria for the non meso-scale eddy? All eddies have the same distance relative to the typhoon center? Is it normalized?

Reply: We could not give you the specific value of the distance to Rmax or typhoon center in Figure 13, for the cold mesoscale eddy area, warm mesoscale eddy area and mesoscale eddy area not indicate one certain eddy area but including all the detected cold mesoscale eddies, warm mesoscale eddies or mesoscale eddies for analyzing the overall change. The non mesoscale eddies area indicates the area without the detected eddies. And we revised the legend in Figure 13 and Figure 15 to avoid misunderstanding. We also give explanation in the manuscript. Besides, we added the distance to the typhoon center in figure 16.

Page 15, line16: The cold mesoscale eddy area includes all the cold mesoscale eddy detected in the study area, which was changing with the generation and dissipation of the cold mesoscale eddy, and the same of the warm mesoscale eddy area.

Comment 10: Figure 14. I’m not sure whether this figure is suitable for showing SST cooling or not. What about SST(D+2)-SST(D-0)?

Reply: We replotted Figure 14 as you suggested and the SST cooling was more obvious.  It can be seen that the reduction of the SST is determined, with the maximum value of -5 °C for Typhoon Chan-hom.

Comment 11: In the section 4, with the figure 15a, there is an abrupt SST increase in both warm and cold eddy at the 10 July 00:00. The authors already pointed out that there is disappearing and generation of eddies. But, I couldn’t follow it due to the limited information (such as eddy map or any?). So, it is confusing with the sentences such as; “within the cold meso~ mainly increased. Conversely, ~ decreased”. I think the variations of SST occurred on 09 July. To reduce misunderstanding, the author needs to rewrite this paragraph for clarity.

Reply: We replotted figure 6 and marked all the mesoscale eddies with number 1-5 in it and make detailed description in the manuscript.

Page 16, line 32: Otherwise, the sudden change of the SST on 00:00 July 9, 2015 and 00:00 July 10, 2015 existed shown in Figure 15 a, this was due to the number 3 warm mesoscale eddy disappeared on July 9, 2015 and number 1 cold mesoscale eddy and number 4 warm mesoscale eddy disappeared on July 10, 2015.

Reviewer 2 Report

Comments and Suggestions for Authors

the paper entitled "The Influence of Typhoon-induced Wave on the Meso-Scale Eddy" is well organized and written well.

there are no special improvements to the paper.

Author Response

Thanks for your positive comments.

Reviewer 3 Report

Comments and Suggestions for Authors

The article provides a comprehensive overview of the current understanding of the chosen topic, discussing the various ways in which typhoon waves can affect the behavior of both cold and warm mesoscale eddies with different radius and eddy activity intensity (EKE). It can be said that the authors effectively explain the mechanisms through which typhoon-induced waves may have an impact on mesoscale eddies and provide rather compelling evidence to support their findings.

However, there are some comments:

1)      One of the strengths of this paper is its use of real-world examples and case studies to illustrate the concepts being discussed. However, the effect of a particular typhoon on the eddies is considered, and the conclusion is formulated in such a way that it seems that these results necessarily apply to all typhoon-induced waves. It might be possible to add the name of the typhoon to the keywords section, and there replace ‘Sea surface temperature’ with the corresponding abbreviation. In addition, it seems the 'mesoscale' variant is more commonly used.

2)      Generally speaking, the paper is clearly written, making it easy to grasp the key concepts and implications to a wide range of readers, from experts in the field to those with a general interest in oceanography. The article includes detailed diagrams and visual aids that further enhance the understanding of the material. Nevertheless, the comparison and conclusions of SSTs from the FVCOM-SWAVE model versus Argos measurements and remote sensing system (REMSS) records are not entirely clear.

        What changes will occur with a change in input parameters?

        Can the simulated SST be validated based on measurements from Argos and REMSS?

        A similar question about the simulated significant wave heights based on measurements from Jason-2 (Figs. 7 and 8).

        The article states that zonal asymmetry of the wave distribution is observed but lacks justification.

3)    Due to the presence of quite a lot of abbreviations and numerical values, the conclusions could be more conveniently represented by bullet points. Some comparisons of the FVCOM-SWAVE with other models could be added to confirm the validity of the data used.

4)    It is necessary to explain the physical mechanism of the cooling and warming effect during the passage of typhoons. The authors write: «For the cold meso-scale eddy, the water in the lower ocean layer was pumped up to the sea surface bringing the heat energy which impeded the cooling effect induced by typhoon. As for the warm meso-scale eddy, the water in the upper ocean layer was transported to the lower ocean layer and formed the downwelling in the center of the eddy which prevented the heat transport 16 from lower ocean layer». It's unclear. If cold water rises, what causes the temperature of the water in the vortex to rise? And vice versa. The explanation with thermal energy does not stand up to any criticism.

5)    The reasoning (lines 20-32 on page 15) has no scientific justification.

Minor comment

The buoy numbers in Figure 4 are poorly readable.

Author Response

We thank the reviewers for their constructive comments that have helped us to improve this manuscript. We provide a point-by-point reply in the document to carefully address these comments and suggestions.

Reply to Reviewer 3

General Comment: The article provides a comprehensive overview of the current understanding of the chosen topic, discussing the various ways in which typhoon waves can affect the behavior of both cold and warm mesoscale eddies with different radius and eddy activity intensity (EKE). It can be said that the authors effectively explain the mechanisms through which typhoon-induced waves may have an impact on mesoscale eddies and provide rather compelling evidence to support their findings.

Reply: Thanks for your suggestions and we added more detail information of mesoscale eddies in this study and rewrote the explanation of the physical mechanism of the cooling and warming effect in typhoon. More detailed revisions were presented in the following.

Comment 1: One of the strengths of this paper is its use of real-world examples and case studies to illustrate the concepts being discussed. However, the effect of a particular typhoon on the eddies is considered, and the conclusion is formulated in such a way that it seems that these results necessarily apply to all typhoon-induced waves. It might be possible to add the name of the typhoon to the keywords section, and there replace ‘Sea surface temperature’ with the corresponding abbreviation. In addition, it seems the 'mesoscale' variant is more commonly used.

Reply: Thanks for your kind suggestion. We added the Typhoon Chan-hom (2015) in the keywords section, replaced the ‘Sea surface temperature’ and ‘meso-scale’ with ‘SST’ and ‘mesoscale’ respectively.

Comment 2: Generally speaking, the paper is clearly written, making it easy to grasp the key concepts and implications to a wide range of readers, from experts in the field to those with a general interest in oceanography. The article includes detailed diagrams and visual aids that further enhance the understanding of the material. Nevertheless, the comparison and conclusions of SSTs from the FVCOM-SWAVE model versus Argos measurements and remote sensing system (REMSS) records are not entirely clear.

–        What changes will occur with a change in input parameters?

–        Can the simulated SST be validated based on measurements from Argos and REMSS?

–        A similar question about the simulated significant wave heights based on measurements from Jason-2 (Figs. 7 and 8).

–        The article states that zonal asymmetry of the wave distribution is observed but lacks justification.

Reply: We added the more details and reference in the manuscript and specific responds was given in point as following.

1) The unstructured triangular used in the simulation determined the area range and spatial resolution. The sponge layer is used to ensure that the radiation condition will also suppress the noise perturbation wave energy reflected back into the computational domain with complex bathymetry at the boundary. The sigma layers are used to determine the vertical resolution and if the surface layer is too thick, the surface parameters such as SST would inaccurate. The in-ternal and internal-external mode are determined by the minimum spatial resolution of the unstructured triangular. The wind force is important for SWAVE simulation especially in the typhoon condition. The tide data, temperature, salinity, and current in the open boundary are important source to simulate current such as Kuroshio.

2) The reliable sea surface temperature data can be also observed using buoys and remote sensing technique, i.e., Argos and remote-sensed data form remote sensing system (REMSS). The primary goal of Argo is to create a systematic global network of profiling floats which provides freely available temperature and salinity data from the upper 2000 m of the ocean with global coverage. The data are available within 24 hours of collection for use in a broad range of applications [8]. The REMSS dataset merges both microwave and infrared data using optimal interpolation. Besides, the product applies a diurnal model to account for day–night differences [9].

Page 2 Line 8: The reliable sea surface temperature data can be also observed using buoys and remote sensing technique, i.e., Argos and remote-sensed data form remote sensing system (REMSS). The primary goal of Argo is to create a systematic global network of profiling floats which provides freely available temperature and salinity data from the upper 2000 m of the ocean with global coverage. The data are available within 24 hours of collection for use in a broad range of applications [8]. The REMSS dataset merges both microwave and infrared data using optimal interpolation. Besides, the product applies a diurnal model to account for day–night differences [9].

Reference

  1. Riser, S.C.; Freeland, H.J.; Roemmich, D.; Wijffels, S.; Troisi, A.; Belbéoch, M.; Gilbert, D.; Xu, J.P.; Pouliquen, S.; Thresher, A.; Le Traon, P.Y.; Maze, G.; Klein, B.; Ravichandran, M.; Grant, F.; Poulain, P.M.; Suga, T.; Lim, B.; Sterl, A.; Sutton, P.; Mork, K.A.; Vélez-Belch, P.J.; Ansorge, I.; King, B.; Turton, J.; Baringer, M.; Jayne, S.R. Fifteen years of ocean observations with the global Argo array. Nat. Clim. Change 2016, 6, 145-153.
  2. Remote Sensing Systems. GHRSST Level 4 MW_OI Global Foundation Sea Surface Temperature Analysis. Ver. 4.0. PO.DAAC, CA, USA, 2019. 2015. Available online: https://doi.org/10.5067/GHMWI-4FR03 (accessed on 19 December 2018).

3) The significant wave height measurements observed by Jason-2 altimeter have been quality controlled [6]. Moreover, the altimeter wave data is also applicable for regional research at basin-scale [7] and there are no available open-access moored buoys there.

Page 2 Line 7: The significant wave height measurements observed by Jason-2 altimeter have been quality controlled [6]. Moreover, the altimeter wave data is also applicable for regional research at basin-scale [7].

Reference

  1. Abdalla, S.; Janssen, Peter A. E. M.; Bidlot, J.R. Jason-2 OGDR Wind and Wave Products: Monitoring, Validation and Assimi-lation. Mar. Geod. 2010, 33, 239-255.
  2. Liu, Q.X.; Babanin, A.V.; Zieger, S.; Young, I.R.; Guan, C.L. Wind and wave climate in the Arctic Ocean as observed by al-timeters. J. Clim. 2016, 29, 7957–7975.

4) We added the typhoon tracks in Figure 9-11. As shown in figure 9, the zonal asymmetry of the wave distribution in the typhoon condition could be identified with the relative high significant wave height on the right side of the typhoon center and relative low significant wave height on the left side of the typhoon center.

Comment 3: Due to the presence of quite a lot of abbreviations and numerical values, the conclusions could be more conveniently represented by bullet points. Some comparisons of the FVCOM-SWAVE with other models could be added to confirm the validity of the data used.

Reply: We added the comparison in the introduction and the conclusions were represented by bullet points.

Page 3 Line 25 Chen et al. [43] evaluated the ADCIRC-SWAN, FVCOM-SWAVE, and SELFE-WWM for simulating extratropical storm and found that the FVCOM-SWAVE could simulate large significant wave height on the coasts concerning the current-wave interaction using a second-order accurate upwind finite-volume scheme.

Comment 4: It is necessary to explain the physical mechanism of the cooling and warming effect during the passage of typhoons. The authors write: «For the cold meso-scale eddy, the water in the lower ocean layer was pumped up to the sea surface bringing the heat energy which impeded the cooling effect induced by typhoon. As for the warm meso-scale eddy, the water in the upper ocean layer was transported to the lower ocean layer and formed the downwelling in the center of the eddy which prevented the heat transport 16 from lower ocean layer». It's unclear. If cold water rises, what causes the temperature of the water in the vortex to rise? And vice versa. The explanation with thermal energy does not stand up to any criticism.

and Comment 5: The reasoning (lines 20-32 on page 15) has no scientific justification.

Reply: We revised this explanation in the section 4.

Page 17 Line 25: The cold mesoscale eddies rotated counter-clockwise, which conflicts with the typhoon and consuming energy, resulting in further weakening the SST cooling induced by typhoon. As for the warm mesoscale eddies, it rotated clockwise, which promotes the development of the typhoon and the SST cooling induced by typhoon. This process was very complex and needed further study. Furthermore, as for the warm meso-scale eddy, the water in the upper ocean layer was transported to the lower ocean layer and formed the downwelling in the center of the eddy which prevented the heat transport from lower ocean layer. Otherwise, the sudden change of the SST on 00:00 July 9, 2015 and 00:00 July 10, 2015 existed shown in Figure 15 a, this was due to the number 3 warm mesoscale eddy disappeared on July 9, 2015 and number 1 cold mesoscale eddy and number 4 warm mesoscale eddy disappeared on July 10, 2015.

Comment 6: The buoy numbers in Figure 4 are poorly readable.

Reply: We replotted Figure 4, the font is larger and the color is changed into white.

Reviewer 4 Report

Comments and Suggestions for Authors

Comments:

1.         Can you elaborate on the methodology used to simulate the wave fields of Typhoon Chan-hom (2015) using the coupled oceanic model (FVCOM-SWAVE) with a nested triangle grid?

2.         How was the forcing wind field determined in the simulation, and what role did the European Centre for Medium-Range Weather Forecasts (ECMWF) reanalysis data and the parametric Holland model (H-E) play in this process?

3.         The validation of simulated oceanic parameters, such as significant wave height (SWH) and sea surface temperature (SST), against Jason-2 altimeter measurements and remote sensing products (REMSS and Argos) is mentioned. Could you explain the significance of the obtained root mean square error (RMSE), correlation coefficient (COR), and scatter index (SI) values?

4.         The abstract mentions the significant zonal asymmetry of the wave distribution along the typhoon track. What factors contribute to this asymmetry, and how does it impact the overall understanding of wave patterns in tropical cyclones?

5.         The calculation of Stokes drift and estimating the contribution of Stokes transport using Ekman–Stokes numbers are discussed. Could you provide more details on how these calculations were performed and their implications for water mixing during a tropical cyclone?

6.         The abstract suggests that the ratio of Stokes transport to the total net transport is > 80% within the maximum wind radius of the typhoon. How does this high ratio influence the dynamics of water transport and mixing in the specified region?

7.         Mesoscale eddies are identified using AVISO's sea-level anomalies (SLA) fusion data. Can you elaborate on the methodology used to detect these eddies and the criteria for distinguishing between warm and cold mesoscale eddies?

8.         The abstract mentions that significant wave heights, Stokes drift, and Stokes transport are higher inside mesoscale eddies than outside. What mechanisms contribute to this difference, and how do these findings enhance our understanding of wave dynamics within mesoscale eddy areas?

9.         The influence of mesoscale eddies on sea surface temperature (SST) is discussed, emphasizing the proportionality to eddy radius and eddy EKE. How are these parameters related, and what insights do they provide into the thermal characteristics of mesoscale eddies?

10.     Could you explain the potential implications of the observed increase in SST within cold mesoscale eddies and the corresponding decrease within warm mesoscale eddies? How does this relate to broader climate and oceanic processes?

11.     The abstract mentions using the hybrid coordinate ocean model (HYCOM) global datasets for open boundary fields. How does including these datasets contribute to the overall accuracy and reliability of the simulation results?

12.     In the study context, how do mesoscale eddies impact the overall distribution of wave energy and current patterns in tropical cyclones?

13.     The abstract notes that the Stokes transport is essential in water mixing during a tropical cyclone. Can you elaborate on the specific mechanisms by which Stokes transport contributes to water mixing and why it is particularly significant in this context?

14.     How do the findings of this study contribute to our understanding of the complex interactions between atmospheric and oceanic processes during tropical cyclones, and what are the potential applications of this knowledge in forecasting and mitigating the impacts of these events?

15.     Considering the observed asymmetry in wave distribution along the typhoon track, how might these findings be applied to improve predictive models for tropical cyclone-induced waves and their impacts on coastal regions?

16.     Many previous studies related to the present study should be considered in the Introduction section, e.g., https://doi.org/10.1016/j.oceaneng.2019.106260; https://doi.org/10.3390/jmse8030217; https://doi.org/10.3390/jmse10101360; https://doi.org/10.3390/jmse11030653

Author Response

We thank the reviewer for their constructive comments that have helped us to improve this manuscript. We provide a point-by-point reply in the document to carefully address these comments and suggestions.

Reply to Reviewer 4

Comment 1: Can you elaborate on the methodology used to simulate the wave fields of Typhoon Chan-hom (2015) using the coupled oceanic model (FVCOM-SWAVE) with a nested triangle grid?

Reply: We elaborated the methodology of FVCOM-SWAVE in the revision (See 2.2 Settings of the FVCOM-SWAVE).

Comment 2: How was the forcing wind field determined in the simulation, and what role did the European Centre for Medium-Range Weather Forecasts (ECMWF) reanalysis data and the parametric Holland model (H-E) play in this process?

Reply: In our previous study (Sheng et al. 2019), we found the maximum wind speeds from ECMWF to be less than those from the JMA best track data. Therefore, we used the parameters from the JMA best track data to build wind fields with the parametric Holland model and ECMWF grid. We chose large values between the ECMWF winds data and simulated Holland winds to obtain H-E wind fields. We collocated the wind and wave data from in-situ buoys around the Zhoushan Islands during the periods of the typhoons Fung-wong in 2014 and Chan-hom in 2015, which we used to validate the H-E winds. The specific methodology has been presented in previous research and will not be repeated in this study.

Reference

Sheng, Y.X.; Shao, W.Z.; Li, S.Q.; Zhang, Y.M.; Yang, H.W.; Zuo, J.C. Evaluation of Typhoon Waves Simulated by WaveWatch-III Model in Shallow Waters Around Zhoushan Islands. J. Ocean U. China 2019, 18, 365-375.

Page 5 Line 40: The maximum wind speeds from ECMWF to be less than those from the JMA best track data. Thus, the H-E winds (the forcing wind field composited from the European Centre for Medium-Range Weather Forecasts (ECMWF) reanalysis data and simulated winds using a parametric Holland model), denoted as H-E, was used as the forcing wind field [42, 44]. While building the H-E winds, the large values between the ECMWF winds data and simulated Holland winds were chosen to obtain H-E wind fields.

Comment 3: The validation of simulated oceanic parameters, such as significant wave height (SWH) and sea surface temperature (SST), against Jason-2 altimeter measurements and remote sensing products (REMSS and Argos) is mentioned. Could you explain the significance of the obtained root mean square error (RMSE), correlation coefficient (COR), and scatter index (SI) values?

Reply: In the manuscript, we used two models to simulate the SWH and SST. For validation, we collected the observations from remote sensing and buoys. The observations obtained from remote sensing product (i.e., REMSS and Jason-2) and buoys (i.e., Argos) are reliable data with high precision which depict the real situation. The statistical parameters used in the paper is explained.

Page 11 Line 12: RMSE, COR and SI were used to analyze the reliability of the simulations. RMSE is used to measure the deviation between simulated values and observed values, and it is sensitive to outliers in the data. COR refers to the degree of correlation between the simulated values and the observed values. When the COR is closer to 1, it indicates a strong positive correlation between the simulated values and the observed values. SI is a measure of dispersion which describes the consistency of the simulated values. A small dispersion coefficient indicates a small degree of data dispersion.

Comment 4: The abstract mentions the significant zonal asymmetry of the wave distribution along the typhoon track. What factors contribute to this asymmetry, and how does it impact the overall understanding of wave patterns in tropical cyclones?

Reply: As indicated in Shi et.al [59], the stronger the typhoon, the less asymmetric the spatial distribution of SWH it causes. Besides, the asymmetry of the wind field and the topography of the shore boundary during typhoons have a significant impact on the spatial distribution of significant wave height.

Page 20 Line 8: As indicated in Shi et.al [59], The stronger the typhoon, the less asymmetric the spatial distribution of SWH it causes. Besides, the asymmetry of the wind field and the topography of the shore boundary during typhoons have a significant impact on the spatial distribution of significant wave height. Thus, reliable TC winds derived from remote-sensed products are anticipated to force the numeric wave model.

  1. Shi, Y.P.; Du, Y.; Chu, X.Q.; Tang, S.L.; Shi, P.; Jiang, X.W. Asymmetric Wave Distributions of Tropical Cyclones Based on CFOSAT Observations. Geophys. Res-Oceans. 2021. https://doi.org/10.1029/2020JC016829.

Comment 5: The calculation of Stokes drift and estimating the contribution of Stokes transport using Ekman–Stokes numbers are discussed. Could you provide more details on how these calculations were performed and their implications for water mixing during a tropical cyclone?

Reply: In this study, these wave parameters including wave number, significant wave height, average period and wave propagation direction were simulated by the FVCOM-SWAVE, the wind speed 10 m above the sea surface was the H-E wind. Furthermore, the large value of Ekman-Stokes number indicated that the contribution of vertical eddy motion in Ekman layer to mixing increases and it might break the ocean layered structure and influence the tracks and intensities of tropical cyclones. And we added the above description in section 2.3.

Page 9 Line 32: In this study, these wave parameters including wave number, significant wave height, average period and wave propagation direction were simulated by the FVCOM-SWAVE and the wind speed 10 m above the sea surface was the H-E wind. Furthermore, the large value of Ekman-Stokes number indicated that the contribution of vertical eddy motion in Ekman layer to mixing increases and it might break the ocean layered structure and influence the tracks and intensities of tropical cyclones.

Comment 6: The abstract suggests that the ratio of Stokes transport to the total net transport is > 80% within the maximum wind radius of the typhoon. How does this high ratio influence the dynamics of water transport and mixing in the specified region?

Reply: The Stokes drift produce water mass movement horizontally and vertically. The horizontal water mass movement could change the temperature, salinity, and other physical properties of the ocean surface layer. Furthermore, the increase of the vertical water mixing might change the ocean layered structure and increase the depth of the mixed layer. We added the above description in section 3.4.

Page 15 Line 9: The Stokes drift was mainly induced by wind and it could be also influenced by Coriolis force and water depth. During the TC, strong wind fields were the main force of Stokes transport which made the Stokes transport dominate the water movement and mixing. The Stokes drift produces water mass movement horizontally and vertically. The horizontal water mass movement could change the temperature, salinity, and other physical properties of the ocean surface layer. Furthermore, the increase of the vertical water mixing might change the ocean layered structure and increase the depth of the mixed layer.

Comment 7: Mesoscale eddies are identified using AVISO's sea-level anomalies (SLA) fusion data. Can you elaborate on the methodology used to detect these eddies and the criteria for distinguishing between warm and cold mesoscale eddies?

Reply: We added the eddy detection criteria and the criteria for distinguishing between warm and cold mesoscale eddies in the revision (See Section 2.4).

Comment 8: The abstract mentions that significant wave heights, Stokes drift, and Stokes transport are higher inside mesoscale eddies than outside. What mechanisms contribute to this difference, and how do these findings enhance our understanding of wave dynamics within mesoscale eddy areas?

Reply: The explanation was given in section 3.5.

Page 16 Line 20: The cold mesoscale eddy area includes all the cold mesoscale eddy detected in the study area, which was changing with the generation and dissipation of the cold mesoscale eddy, and the same of the warm mesoscale eddy area. The values of the significant wave heights, Stokes drift and Stokes transport inside the cold mesoscale eddies areas were all greater than which inside the warm mesoscale eddies areas. As the cold eddies were cyclonic eddies, this kind of eddies pumped up the water in the lower ocean layer to the sea surface, forming the upwelling and increasing the significant wave heights. While the significant wave heights increased, the Stokes drift and Stokes transport increased. As for the warm eddies, this kind of eddies were anti-cyclonic eddies and performed conversely compared to the cold eddies.

Comment 9: The influence of mesoscale eddies on sea surface temperature (SST) is discussed, emphasizing the proportionality to eddy radius and eddy EKE. How are these parameters related, and what insights do they provide into the thermal characteristics of mesoscale eddies?

Reply: We added the explanation in section 4.

Page 17 Line 25: The cold mesoscale eddies rotated counter-clockwise, which conflicts with the typhoon and consuming energy, resulting in further weakening the SST cooling induced by typhoon. As for the warm mesoscale eddies, it rotated clockwise, which promotes the development of the typhoon and the SST cooling induced by typhoon. This process was very complex and needed further study. Furthermore, as for the warm meso-scale eddy, the water in the upper ocean layer was transported to the lower ocean layer and formed the downwelling in the center of the eddy which prevented the heat transport from lower ocean layer. Otherwise, the sudden change of the SST on 00:00 July 9, 2015 and 00:00 July 10, 2015 existed shown in Figure 15 a, this was due to the number 3 warm mesoscale eddy disappeared on July 9, 2015 and number 1 cold mesoscale eddy and number 4 warm mesoscale eddy disappeared on July 10, 2015.

Comment 10: Could you explain the potential implications of the observed increase in SST within cold mesoscale eddies and the corresponding decrease within warm mesoscale eddies? How does this relate to broader climate and oceanic processes?

Reply: We added the explanation in the discussion as following:

Page 18 Line 10: Mesoscale eddies with large radius might have stronger structures while large eddy EKE indicated large vorticity intensity and strong wind force. Thus, mesoscale eddies with large radius and EKEs had more significant effect on the SST cooling induced by typhoon. This phenomenon might affect the distribution, growth and stability of the ecological chain of phytoplankton and animals. In the aspect on a global scale, the regulation of SST by mesoscale eddies may affect large-scale ocean currents and heat distribution.

Comment 11: The abstract mentions using the hybrid coordinate ocean model (HYCOM) global datasets for open boundary fields. How does including these datasets contribute to the overall accuracy and reliability of the simulation results?

Reply: The open boundary allows the momentum or mass to be radiated out of or flow into the computational domain with temperature, salinity, and current. The specified tidal elevation at the open boundary can guarantee to radiate the fastest surface gravity wave energy out of the computational domain and to ensure the mass conservation throughout the water column. And these are important sources to simulate circulation such as Kuroshio. We added the explanation in section 2.2.

Comment 12: In the study context, how do mesoscale eddies impact the overall distribution of wave energy and current patterns in tropical cyclones?

Reply: Mesoscale eddies could affect the wave energy exchange and current patterns in the tropical cyclones condition by changing the vertical mixing of seawater especially in the upper ocean layer. Otherwise, the interaction of mesoscale eddies and topography such as submarine ridges can also altering the path and velocity of ocean currents. These changes may affect the distribution of ocean currents throughout the entire tropical cyclone region. And we added the explanation in section 3.5.

Page 15 Line 22: The mesoscale eddies are common marine phenomenon. Mesoscale eddies could affect the wave energy exchange and current patterns under the tropical cyclones condition by changing the vertical mixing of seawater, especially in the upper ocean layer. Otherwise, the interaction of mesoscale eddies and topography such as submarine ridges can also altering the path and velocity of ocean currents. These changes may affect the distribution of ocean currents throughout the entire tropical cyclone region.

Comment 13: The abstract notes that the Stokes transport is essential in water mixing during a tropical cyclone. Can you elaborate on the specific mechanisms by which Stokes transport contributes to water mixing and why it is particularly significant in this context?

Reply: The Stokes drift is mainly induced by wind and it could be also influenced by Coriolis force and water depth. The Stokes drift produce water mass movement horizontally and vertically to enhance the interaction between water mass. During tropical cyclones, strong wind fields are the main force of Stokes transport which makes the Stokes transport dominate the water movement and mixing.

Comment 14: How do the findings of this study contribute to our understanding of the complex interactions between atmospheric and oceanic processes during tropical cyclones, and what are the potential applications of this knowledge in forecasting and mitigating the impacts of these events?

Reply: With this study, the interaction of tropical cyclone and mesoscale eddies was presented, and the transport contribution of stokes drift during tropical cyclone was exposed. These understandings of Stokes transport and mesoscale eddies can be incorporated into tropical cyclone simulation models to improve the prediction accuracy. Furthermore, it helps improve meteorological forecasting and risk management. And we added it in the section of conclusions.

Page 20 Line 29: These understandings of Stokes transport and mesoscale eddies can be incorporated into TC simulation models to improve the prediction accuracy. Furthermore, it helps improve meteorological forecasting and risk management during TCs.

Comment 15: Considering the observed asymmetry in wave distribution along the typhoon track, how might these findings be applied to improve predictive models for tropical cyclone-induced waves and their impacts on coastal regions?

Reply: As mentioned in reply 4, the typhoon strength, the asymmetry of the wind field and the topography of the shore boundary have a significant impact on the spatial distribution of significant wave height. Thus, the accurate wind forcing and topography with fine resolution could improve predictive models. In particular, the remote-sensed winds are anticipated to force the numeric wave model.

Page 20 Line 8: As indicated in Shi et.al [59], The stronger the typhoon, the less asymmetric the spatial distribution of SWH it causes. Besides, the asymmetry of the wind field and the topography of the shore boundary during typhoons have a significant impact on the spatial distribution of significant wave height. Thus, reliable TC winds derived from remote-sensed products are anticipated to force the numeric wave model.

Comment 16: Many previous studies related to the present study should be considered in the Introduction section, e.g., https://doi.org/10.1016/j.oceaneng.2019.106260; https://doi.org/10.3390/jmse8030217; https://doi.org/10.3390/jmse10101360; https://doi.org/10.3390/jmse11030653

Reply: We added the description of fusion wind field in the typhoon condition and the four references in the manuscript.

Page 2 Line 29: The forcing wind field is an importance source while simulating the wave filed, and the extreme huge wave induced by typhoon was severely underestimated if only the reanaly-sis wind products (i.e., the Climate Forecast System Reanalysis (CFSR), ERA-Interim (ERA) and Cross-Calibrated Multi-Platform version 2 (CCMPV2)) were utilized. The hybrid wind fields combined of parametric typhoon models (i.e., the Holland model) and the reanaly-sis wind products can improve the performance of wave models [17,18,19,20].

Reference

  1. Chen, W.B.; Chen, H.; Hsiao, S.C.; Chang, C.H.; Lin, L.Y. Wind forcing effect on hindcasting of typhoon-driven extreme waves. Ocean Eng. 2019, 188, 106260.
  2. Chen, W.B. Typhoon Wave Simulation Responses to Various Reanalysis Wind Fields and Computational Domain Sizes. J. Mar. Sci. Eng. 2022, 10, 1360.
  3. Hsiao, S.C.; Chen, H.; Wu, H.L.; Chen, W.B.; Chang, C.H.; Guo, W.D.; Chen, Y.M.; Lin, L.Y. Numerical Simulation of Large Wave Heights from Super Typhoon Nepartak (2016) in the Eastern Waters of Taiwan. J. Mar. Sci. Eng. 2020, 8, 217.
  4. Hsiao, S.C.; Wu, H.L.; Chen, W.B. Study of the Optimal Grid Resolution and Effect of Wave–Wave Interaction during Simu-lation of Extreme Waves Induced by Three Ensuing Typhoons. J. Mar. Sci. Eng. 2023, 11, 653.

Round 2

Reviewer 1 Report

Comments and Suggestions for Authors

The authors revised the manuscript in response to the the reviewer's comments. The revised version of manuscript is now suitable for publication.

Author Response

We appreciate these positive comments.

Reviewer 3 Report

Comments and Suggestions for Authors

1)      The authors should write mesoscale instead of meso-scale.

https://en.wikipedia.org/wiki/Mesoscale_meteorology

2)      The term ‘the indeterminacy eddy center’ is unsuccessful (incomprehensible). Why ‘the eddy center’?

3)      P. 11, L. 10. ‘Stream function’ or ‘flow function’? It is not clear from the text.

Author Response

1)      The authors should write mesoscale instead of meso-scale.

https://en.wikipedia.org/wiki/Mesoscale_meteorology

 Reply: corrected in the title.

2)      The term ‘the indeterminacy eddy center’ is unsuccessful (incomprehensible). Why ‘the eddy center’?

Reply: we use ‘the eddy center’ in the revision.

3)      P. 11, L. 10. ‘Stream function’ or ‘flow function’? It is not clear from the text.

Reply: we use ‘stream function’ in the revision.

Reviewer 4 Report

Comments and Suggestions for Authors

The authors have comprehensively addressed all the comments raised by the reviewer. The current version of the manuscript is now acceptable for publication.

Author Response

We appreciate this positive comment.